# Improving Diffusion Models for Scene Text Editing with Dual Encoders

**Jiabao Ji**[1][*]    **Guanhua Zhang**[1][*]

**Zhaowen Wang**[2]   **Bairu Hou**[1]   **Zhifei Zhang**[2]   **Brian Price**[2]   **Shiyu Chang**[1]

[1] *University of California, Santa Barbara*   [2] *Adobe Research*

[*] *Equal contribution. Work done while Guanhua was affiliated with UCSB.*

*Correspondance to <jiabaoji@ucsb.edu> <guanhua.zhang@tuebingen.mpg.de>.*

**Reviewed on OpenReview:** *https://openreview.net/forum?id=yL15ys5swq*

## Abstract

Scene text editing is a challenging task that involves modifying or inserting specified texts in an image while maintaining its natural and realistic appearance. Most previous approaches to this task rely on style-transfer models that crop out text regions and feed them into image transfer models, such as GANs. However, these methods are limited in their ability to change text style and are unable to insert texts into images. Recent advances in diffusion models have shown promise in overcoming these limitations with text-conditional image editing. However, our empirical analysis reveals that state-of-the-art diffusion models struggle with rendering correct text and controlling text style. To address these problems, we propose DIFFSTE to improve pre-trained diffusion models with a dual encoder design, which includes a character encoder for better text legibility and an instruction encoder for better style control. An instruction tuning framework is introduced to train our model to learn the mapping from the text instruction to the corresponding image with either the specified style or the style of the surrounding texts in the background. Such a training method further brings our method the zero-shot generalization ability to the following three scenarios: generating text with unseen font variation, *e.g.,* italic and bold, mixing different fonts to construct a new font, and using more relaxed forms of natural language as the instructions to guide the generation task. We evaluate our approach on five datasets and demonstrate its superior performance in terms of text correctness, image naturalness, and style controllability. Our code is publicly available at `https://github.com/UCSB-NLP-Chang/DiffSTE`.

## 1 Introduction

Scene text editing has gained significant attention in recent years due to its practical applications in various fields, including text image synthesis Karacan et al. (2016); Qin et al. (2021); Zhan et al. (2018), styled text transfer Atarsaikhan et al. (2017); Azadi et al. (2018); Zhang et al. (2018), and augmented reality translation Cao et al. (2023); Du et al. (2011); Fragoso et al. (2011). The task involves modifying or inserting specified text in an image while maintaining its natural and realistic appearance Huang et al. (2022); Shi et al. (2022); Wu et al. (2019). Most previous approaches have formulated the problem as a style transfer task using generative models such as GANs Lee et al. (2021); Park et al. (2021); Qu et al. (2022); Roy et al. (2020); Wu et al. (2019); Yang et al. (2020).

Specifically, these methods rely on a reference image with the target style, *e.g.*, cropped out text region that needs to be modified. The method then transfers a rendered text in the desired spelling to match the style and the background of the reference. However, these methods are limited in their ability to generate text in arbitrary styles (*e.g.*, font and color) or at arbitrary locations. Additionally, the process of cropping, transferring style, and then replacing back often leads to less natural-looking results. Figure 1.(c) illustrates an image generated by MOSTEL Qu et al. (2022), a state-of-the-art GAN-based scene text editing model.

The edited part of the image exhibits discordance with the surrounding areas with distinct boundaries and messy distortions.

On the other hand, recent advances in diffusion models have shown promise for overcoming these limitations and creating photo-realistic images Balaji et al. (2022); Ho et al. (2020); Saharia et al. (2022); Song et al. (2020). Furthermore, by incorporating a text encoder, diffusion models can be adapted to generate natural images following text instructions, making them well-suited for the task of image editing Ramesh et al. (2022); Rombach et al. (2022). Despite the remarkable success of diffusion models in general image editing, our empirical study reveals that they still struggle with generating accurate texts with specified styles for scene text editing. For instance, Figure 1.(d) shows an example of the text generated by stable-diffusion-inpainting (SD) Rombach et al. (2022), a state-of-the-art text-conditional inpainting model. We feed the model with the masked image and a text instruction, requiring the model to complete the masked region. Although the resulting image shows better naturalness, neither the spelling of the generated text is correct nor the specified style is well followed by the model.

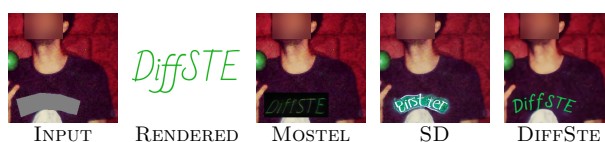

INPUT    RENDERED    MOSTEL    SD    DIFFSTE

Figure 1: Text editing results by GAN-based MOS-TEL, diffusion model SD, and our method DIFFSTE. A reference RENDERED image with desired content and style is fed to MOSTEL as style guidance. SD and our method are prompted by instruction *Write a grass-colored word "DiffSTE" in BadScript font.*

Motivated by this, in this paper, we aim to improve the pre-trained diffusion models, especially SD, in their scene text editing ability. In particular, we identify two main issues as the cause of the diffusion model's failure: ① although the target text is fed as a part of the input to the text encoder of SD, it cannot encode the character-level information of the target text, making it challenging to accurately map the spelling to appropriate visual appearance; ② the models are unable to understand the language instructions for a style-specified generation, which leads to the incorrect font or color generation. To address these challenges, we propose a novel model for scene text editing called DIFFSTE, which incorporates a dual encoder design comprising of an instruction encoder and a character encoder. Specifically, DIFFSTE is built upon SD, and the instruction encoder is inherited from the original SD's text encoder. The input to this encoder describes the target text and its desired style, which is then utilized in cross-attention to guide the generation process. On the other hand, the character encoder is an add-on module that only takes the character-tokenized target text. The encoded character embeddings are then attended by the image encodings and further aggregated together with the attended result from the instruction encoder side. Having direct access to the characters equips the model with the capability of explicit spelling checks and the awareness of target text length.

To make both encoders understand the scene text editing task, we further introduce an instruction-tuning framework for training our model. Instruction tuning is a popular technique in the natural language processing (NLP) domain Gupta et al. (2022); Mishra et al. (2022); Sanh et al. (2021); Wei et al. (2021); Honovich et al. (2022); Wang et al. (2022); Xu et al. (2022); Zhong et al. (2022); Zhou et al. (2022). It aims to teach foundation language models to perform specific *tasks* based on the corresponding *task instructions*. By doing so, models can achieve better performance on both training and unseen tasks with appropriate instructions Chung et al. (2022); Chen et al. (2022); Gu et al. (2022); Jang et al. (2023); Muennighoff et al. (2022); Brooks et al. (2022). Similarly, our goal is to teach DIFFSTE to capture the mapping from the text instruction to scene text generation. To achieve this, we create a synthetic training dataset, where each training example includes three components: ① a text instruction that describes the desired target text, ② a masked region surrounded by other scene texts in a similar style indicating the location in the image where the text should be generated, and ③ a ground truth reference image. The text instructions are generated using a fixed natural language rule that can be grouped into four main categories: specifying both color and font, specifying color only, specifying font only, and providing no style specification. If a text instruction is missing either color or font information or both, the scene text in the surrounding area of the image can be used to infer the missing information.

Our model is trained to generate the ground-truth scene text by minimizing a mean square error (MSE) loss conditioned on the provided instructions and the unmasked image region using both synthetic and real-world datasets. This enables our model to achieve both spelling accuracy and good style control. An example of

the generated image can be found in Figure 1.(e). Even more surprisingly, our model performs well on test instructions that were significantly different from the ones used for training, demonstrating its zero-shot generalization ability. Specifically, our model generalizes well to the following three scenarios: generating text with unseen font variation, *e.g.,* italic and bold, mixing different fonts to construct a new font, and using more relaxed forms of natural language as the instructions to guide the generation task. Figure 2 showcases the new font style created by DIFFSTE, which mixes the fonts *PressStart2P* and *MoonDance.*

We compared our approach with state-of-the-art baselines on five datasets, considering both style-free and style-conditional text editing scenarios. Our method showed superior performance in terms of text correctness, image naturalness, and style control ability, as evaluated quantitatively and subjectively. For instance, under the style-free text editing setting, DIFFSTE achieves an average improvement of 64.0% on `OCR` accuracy compared to the most competitive baseline. Additionally, our approach maintained high image naturalness, as demonstrated by 28.0% more preference over other baseline methods in human evaluations. In the style-conditional scenario, our method shows 69.2% and 26.8% more preference compared to other diffusion baselines in terms of font correctness and color correctness, with consistent improvements in text correctness and naturalness as observed in the style-free case.

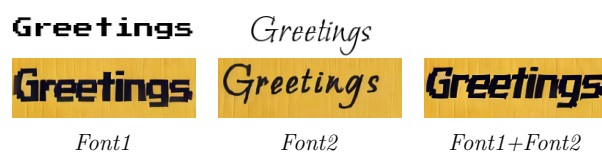

Figure 2: Generating text in new style by blending two existing fonts. The first row shows the texts rendered with real fonts, and the second row shows our generation results with instructions *Write "Greetings" in font:* FONT, where FONT is *PressStart2P*, *MoonDance* and *PressStart2P and MoonDance*, respectively.

## 2    Related Work

**Scene text editing**    GAN-based style transfer methods have been widely used for the task of scene text editing Huang et al. (2022); Kong et al. (2022b); Lee et al. (2021); Roy et al. (2020); Shimoda et al. (2021); Yang et al. (2020); Zhan et al. (2019). These works accomplish the task by transferring the text style in a reference image to the rendered target text image. STEFANN Roy et al. (2020) edits scene text at the character level with a font-adaptive neural network for extracting font structure and a color-preserving model for extracting text color. SRNet Wu et al. (2019) breaks down scene text editing into two sub-tasks, which involve extracting the image background and text spatial alignment, and a fusion model is then used to generate the target text with the extracted information. Mostel Qu et al. (2022) further incorporates additional stroke-level information to improve scene text editing performance. To reduce the dependence of paired synthetic data with source style images, and target style images, TextstyleBrush Krishnan et al. (2023) proposes to disentangle the text appearance into content and style vectors in a self-supervised manner, which allows the utilization of real-world text images. Another recent work, SSTE Su et al. (2023), proposes to embed the text styles in the latent feature space, thus allowing users to control the text style, such as text rotation, text color, and font via latent space editing similar to StyleGAN Karras et al. (2019). Despite the reasonable performances, these methods are limited in generating text in arbitrary styles and locations and often result in less natural-looking images.

**Diffusion models**    Recent advances in diffusion models have achieved significant success in image editing Balaji et al. (2022); Brack et al. (2023); Ho et al. (2020); Li et al. (2023); Nichol et al. (2021); Nichol & Dhariwal (2021); Ryu & Ye (2022); Wu & De la Torre (2022); Wu et al. (2022); Xie et al. (2022). Stable diffusion Rombach et al. (2022) is one of the state-of-the-art diffusion models, which enables effective text-conditional image generation by first compressing images into a low-dimensional space with an autoencoder and then leveraging text encodings with a cross-attention layer. The model could be easily adapted for various tasks like image inpainting and image editing conditioned on texts. However, it has been observed that the diffusion models exhibit poor visual text generation performance and are often susceptible to incorrect text generation Ramesh et al. (2022); Saharia et al. (2022). There is a limited amount of research focusing on improving the text-generation capabilities of diffusion models. A recent study has sought to improve

the generation of visual text images by training a character-aware diffusion model conditioned on a large character-aware text encoder Liu et al. (2022). However, this work differs from our method in the application, as they concentrate on text-to-image generation, while our work focuses on editing texts within an image. In contrast to their single-encoder architecture, our approach incorporates dual encoders to provide character and style encoding for the target text, respectively. Some other concurrent works, such as ControlNet Zhang & Agrawala (2023), DiffUTE Chen et al. (2023a), and TextDiffuser Chen et al. (2023b), have demonstrated exceptional performance in image editing by providing the model with references such as canny edge images and character segmentation maps. However, such fine-grained text reference is challenging to apply for scene text editing. Moreover, the reference can only be used to control the text content, while the text style is not considered. In contrast, our method only needs natural language instruction to control the generated text and style, enjoying better scalability and flexibility.

## 3 Methodology

The proposed DIFFSTE aims to improve the pre-trained diffusion models in their scene text editing ability. We specifically use the SD, a state-of-the-art diffusion model, as our backbone, but the proposed methodology is general to other text-to-image diffusion model architectures. The key architectural innovation of DIFFSTE is to augment SD with an additional character-level encoder to form a dual-encoder structure. We will first elaborate on our model architecture with the dual-encoder design. An illustration can be seen in Figure 3. Next, we will describe our instruction tuning framework for training our model.

### 3.1 The Dual-encoder Design

Our goal is to equip pre-trained diffusion model SD with the ability to accurately capture the spelling of the target text while understanding the provided instructions for generating the target text in the desired style and font. To achieve this, we consider a dual-encoder architecture in DIFFSTE, which consists of an instruction encoder $\text{ENC}_{\text{inst}}$ and a character encoder $\text{ENC}_{\text{char}}$. The instruction encoder $\text{ENC}_{\text{inst}}$ is inherited from the text encoder of SD, namely, a CLIP Radford et al. (2021) text encoder. It is expected to encode the desired style information for target text generation with the proper instruction as input. In contrast, the character encoder $\text{ENC}_{\text{char}}$ is a new add-on module to SD that takes only the target text to be generated tokenized at the character level as input, providing DIFFSTE the direct access to the correct spelling.

To incorporate the information from both encoders, we modify only the cross-attention module from SD. Specifically, unlike SD, which only computes the cross-attention from the instruction encoder side, we construct the cross-attention layer by first separately attending to the embeddings from the instruction and character levels and then aggregating the attended results using a simple

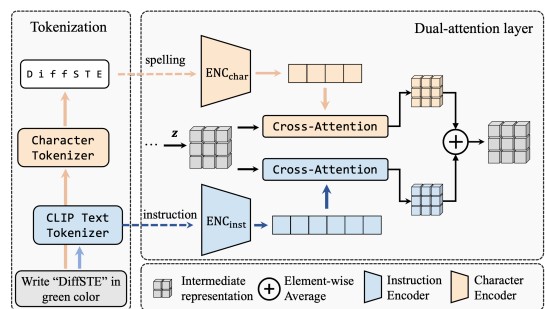

Figure 3: Model structure of DIFFSTE with dual-encoder design. The input text instruction and the target text spelling are processed by instruction encoder $\text{ENC}_{\text{inst}}$ and character encoder $\text{ENC}_{\text{char}}$, respectively. Both encodings are attended to the intermediate hidden representation $z$ from the previous layer through cross-attention. The results are averaged to guide image generation.

average. As shown in Figure 3, both encodings are used to attend the intermediate hidden representation $z$ (output of the previous layer) in the UNet components during visual denoising. The averaged result is then fed into the following UNet layer. By applying this modified cross-attention at every denoising layer, the generated clean image at the end of the diffusion process is expected to be natural, with correct spelling, and consistent with the language instruction.

### 3.2 Instruction Tuning

We adopt an instruction tuning framework to train our model to understand the scene text editing task. Similar to the instruction tuning technique in the NLP area, where foundation models are taught to follow task instructions to perform the corresponding task, our goal is to teach our dual-encoder DIFFSTE model to accurately map text instructions to the corresponding visual appearance. To achieve this goal, we first introduce a synthetic dataset using an automatic rendering engine. Each constructed example contains ① a language instruction that describes the desired target text, ② a masked image indicating where scene text editing should be performed, and ③ a corresponding ground truth image. We have also included existing real-world scene text editing datasets in our instruction tuning process to enable DIFFSTE to generalize to a wide range of real-world backgrounds and scenarios. In the following, we provide a detailed explanation of how we generate the synthetic dataset and how we convert the existing dataset to a similar format so it can be utilized in our instruction tuning pipeline. Once these datasets are appropriately prepared, DIFFSTE is trained to minimize the MSE loss between the ground truth image and the generated image conditioned on the text instruction and the masked image.

**Synthetic dataset creation**   We begin by randomly selecting a few English words or numerical numbers and pairing each of them with a color and font style. We then utilize a publicly available synthesizing engine, Synthtiger Yim et al. (2021), to render each text using its corresponding specified style. Afterward, we apply rotation and perspective transformation to these rendered scene texts Qu et al. (2022) and place them in non-overlapping locations on a background image selected at random from the SynthText Gupta et al. (2016) project. After constructing the image, one random word in the image is masked out to create a masked input, and the corresponding text and style information is utilized to generate task instructions. In particular, we consider two settings to construct the instructions, which are style-free and style-specific. As the name suggests, for the former case, the instruction only provides the specific text we want the model to generate without specifying the style. In this scenario, we ensure that the style of the surrounding scene texts to the masked one is the same and consistent with the masked ground truth. Here, our model is required to follow the style of other texts in the surrounding area to generate the target text correctly and naturally. For the style-specific setting, we consider three different cases, namely generating the scene text with font specification only, color specification only, and both font and color specification. Similarly, in the first two partial style instruction cases, the surrounding texts in the masked image always conform to the missing style, so the model could infer the correct style from the background to recover the ground truth image. On the other hand, in the last case, the surrounding texts are not necessarily consistent with the ground truth style of the target text, so the model would have to follow text instructions to generate the correct style.

There are four categories of instructions in total, and we generated these instructions for our synthetic dataset using a simple fixed rule. For the style-free case, our instruction is created simply by using the rule *Write "TEXT"*, where TEXT is the ground truth masked word. For the case where the style is specified, we construct the instruction following the rule *Write "TEXT" in color: COLOR and font: FONT*. Furthermore, for the partial style-missing case, we modify the previous full instruction rule and remove the corresponding missing information. Although our instructions are created following a very simple rule, we will later show that the model tuned with these instructions can generalize well to other free-form language instructions.

**Real-world dataset utilization**   Commonly used real-world scene editing datasets such as `ArT` Chng et al. (2019), `COCOText` Gomez et al. (2017), and `TextOCR` Singh et al. (2021) only provide the bounding box locations of scene texts in images without text style information. Therefore, we only generate style-free instructions to pair with real-world images for instruction tuning.

## 4 Experiments

### 4.1 Experiment Setup

**Datasets**   As described in Section 3.2, we collect 1.3M examples by combining the synthetic dataset (`Synthetic`) and three real-world datasets (`ArT`Chng et al. (2019), `COCOText`Gomez et al. (2017), and `TextOCR` Singh et al. (2021)) for instruction tuning. For the`Synthetic` dataset, we randomly pick up

100 font families from the google fonts library[1] and 954 XKCD colors[2] for text rendering. We randomly select 200 images from each dataset for validation and 1000 images for testing. Additionally, we test our method on another real-world dataset, `ICDAR13` Karatzas et al. (2015), to assess the model's generalization ability. Different from the training examples described in Section 3.2, the language instruction for each test example is constructed with a random target text, which is different from the text in the original image. In this way, we focus on the more realistic text editing task and measure the result quality with human evaluation. All the images are cropped/resized to 256×256 resolution as model inputs. More details can be found in Appendix B.

**Baselines** We compare DiffSTE with state-of-the-art GAN-based style-transfer methods and diffusion models described below. The GAN-based methods include SRNet Wu et al. (2019) and Mostel Kong et al. (2022a). The training of SRNet requires "parallel" image pairs with different texts appearing at the same location and background, which is not available for real-world datasets, so we fine-tune the released model of SRNet only on our synthetic dataset. We use the original model for Mostel as we empirically find that fine-tuning on our data does not improve its performance. For the diffusion model baselines, we include pre-trained stable-diffusion-inpainting (SD) and stable-diffusion-2-inpainting (SD2). These two models are further fine-tuned by instruction tuning in the same way as our method, as described in Section 3.2. The resulting models are termed as SD-FT and SD2-FT. More implementation details could be found in Appendix B.

**Evaluation and metrics** Our evaluations are conducted under style-free and style-conditional settings. Style-free generation focuses on the correctness and naturalness of the generated texts, with the language instruction only describing the target text content. All diffusion-based methods are fed with the text instruction and the masked image. On the other hand, as GAN-based models SRNet and Mostel cannot take text instructions as input, we feed them with the cropped-out text region for editing in the original image and a rendered text image with the target text. Their outputs are filled into the masked image as the final generation. Note that the GAN-based methods enjoy extra advantages compared with ours as they have access to the style of the original text. This makes it easier for them to generate natural-looking images by simply following the original text style. In contrast, our method has to infer the style of the target text from other surrounding texts in the masked image for a natural generation.

For the style-conditional generation, we require all methods to generate the target text following a specified style. In this setting, the text instruction describes the target text with a random color and font style specified. The diffusion-based methods are fed with the style-specified text instruction and the masked image. However, the GAN-based style transfer models, SRNet and Mostel, do not support style specification by texts. Thus, we synthesize a style reference image of a random text using the specified font and color as well as a rendered image with target text. Both images are fed into GAN-based models as input, and the resulting output is filled into the masked image as the final generation. Note that the style reference images can be different in synthetic and real-world testing sets. For the synthetic testing set, the synthesized reference image contains the same background as the original image. For the real-world testing sets, a pure-colored background is used as we can not synthesize an image with texts in the specified target style on the same background as the original image.

We evaluate the effectiveness of our method from three aspects: ① text correctness, ② image naturalness, and ③ style correctness. For ① text correctness, we report the _OCR_ accuracy, which is calculated as the exact match rate between the target text and the recognized text from the generation by a pre-trained recognition model Fang et al. (2021). Human evaluation of the text accuracy is also reported, denoted as _Cor_. For ② image naturalness, we ask the human evaluator to compare the generated images by a certain baseline and our method DiffSTE, and the vote rate gap (vote of baseline minus vote of DiffSTE) is reported as naturalness score _Nat_. For ③ style correctness, under the style-specific evaluation setting, human evaluators are presented with a rendered reference image of the target text in the specified style and are asked to vote between a baseline and our method _w.r.t._ font correctness and color correctness of the generated images. The resulting vote gaps are reported as _Font_ and _Color_, respectively. All human evaluation results are conducted on 50 randomly selected images for each of the five datasets. More details could be found in Appendix A.

---

[1] https://fonts.google.com
[2] https://xkcd.com/color/rgb/

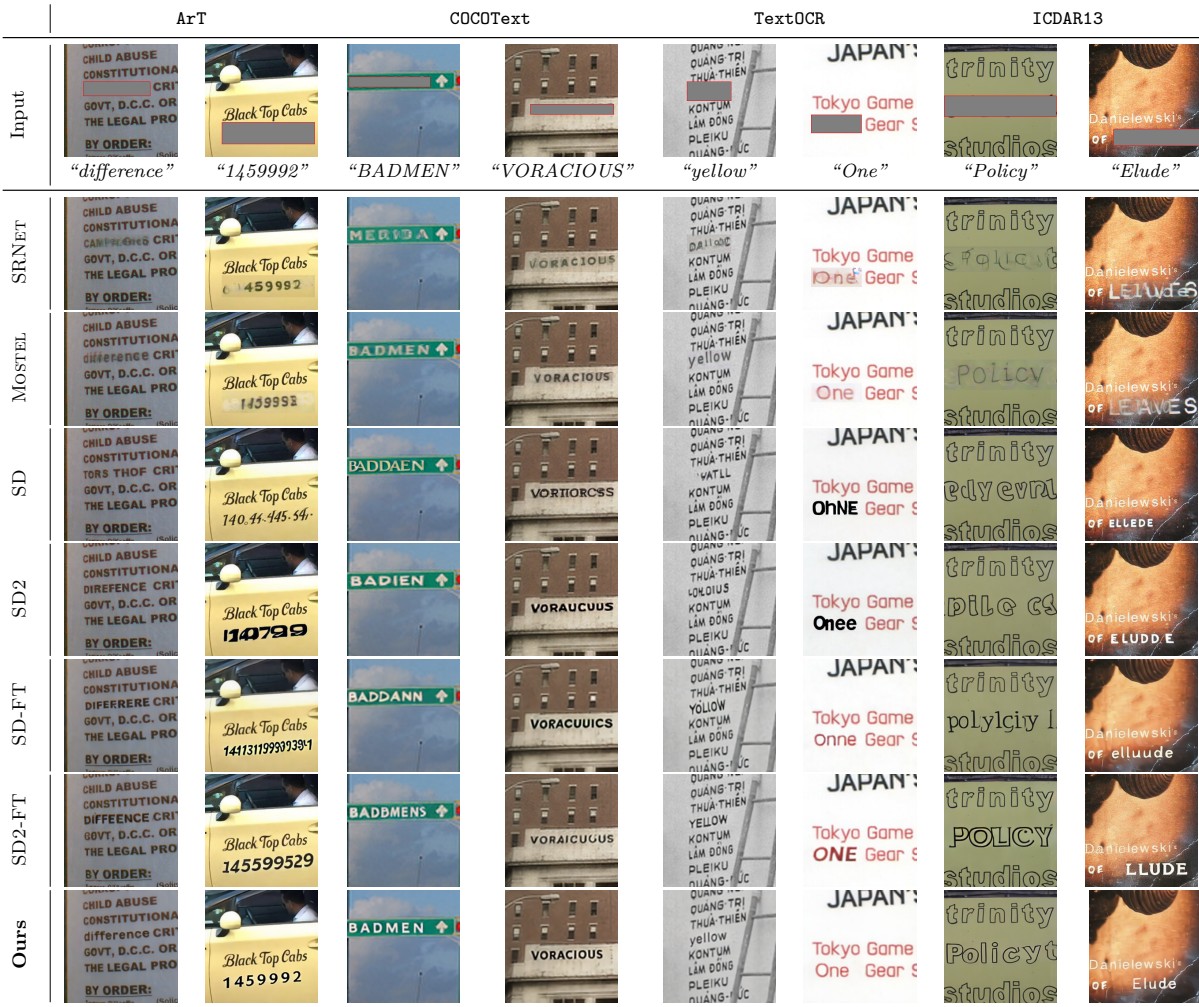

Figure 4: Examples of style-free text editing results on four real-world datasets.

## 4.2 Experiment Results

**Style-free generation**  The quantitative results for style-free generation are shown in Table 1. We observe that our method DIFFSTE consistently achieves better text correctness and image naturalness than other baselines over all datasets. For example, DIFFSTE improves average *OCR* accuracy and human-evaluated text correctness (*Cor*) by 64.0% and 21.7% compared with the most competitive baseline MOSTEL. For human evaluated naturalness (*Nat*), DIFFSTE outperforms all other baselines by at least 28% more vote percentage on average. Besides, the superiority of our method over other fine-tuned diffusion-based methods SD-FT and SD2-FT have demonstrated the effectiveness of our newly introduced character encoder.

We present the qualitative results in Figure 4 and Appendix D.1. We emphasize the effectiveness of DIFFSTE as follows. First, DIFFSTE generates texts strictly following the given instruction, which is consistent with the superior performance of DIFFSTE in terms of text correctness in quantitative evaluation. In comparison, other diffusion-based baselines are prone to generating wrong spellings, while the GAN-based methods tend to generate blurry and ambiguous texts and thus get low text correctness. For example, in the `ArT` dataset with the target text "difference", with our method perfectly generating the correct text, SD-FT generates "DIFEFRERE" while the generation of MOSTEL is hard to read due to the blur. Second, our method generates more natural images than other baselines. For example, in the `TextOCR` dataset with "One" target text, the "One" generated by DIFFSTE conforms to the font and color of other texts in the context. By contrast, GAN-based methods SRNET and MOSTEL also copy the text style from context and generate red "One", but the edited region is incoherent with the background with a clear boundary.

| Method | Synthetic | | | ArT | | | COCOText | | | TextOCR | | | ICDAR13 | | | Average | | |
|---|---|---|---|---|---|---|---|---|---|---|---|---|---|---|---|---|---|---|
| | OCR↑ | Cor↑ | Nat↑ | OCR↑ | Cor↑ | Nat↑ | OCR↑ | Cor↑ | Nat↑ | OCR↑ | Cor↑ | Nat↑ | OCR↑ | Cor↑ | Nat↑ | OCR↑ | Cor↑ | Nat↑ |
| **Style-free generation** | | | | | | | | | | | | | | | | | | |
| SRNet | 50.74 | 44 | -28 | 30.45 | 42 | -16 | 26.87 | 84 | -68 | 31.14 | 42 | -40 | 30.13 | 46 | -52 | 33.87 | 51.6 | -40.8 |
| Mostel | 71.51 | 80 | -12 | 60.99 | 70 | -20 | 24.11 | 84 | -40 | 46.11 | 56 | -52 | 52.23 | 60 | -20 | 50.99 | 70.0 | -28.8 |
| SD | 3.12 | 6 | -40 | 5.39 | 6 | 0 | 12.46 | 10 | -16 | 8.22 | 4 | -40 | 4.56 | 6 | -44 | 6.75 | 6.4 | -28.0 |
| SD2 | 4.61 | 8 | -32 | 7.28 | 16 | -20 | 19.34 | 14 | -36 | 11.88 | 14 | -30 | 9.13 | 8 | -48 | 10.45 | 12.0 | -33.2 |
| SD-FT | 29.51 | 34 | -64 | 32.08 | 34 | -40 | 51.62 | 60 | -4 | 48.91 | 44 | -48 | 25.29 | 30 | -24 | 37.48 | 40.4 | -36.0 |
| SD2-FT | 37.53 | 46 | -40 | 46.17 | 44 | -48 | 56.13 | 50 | -32 | 60.82 | 56 | -24 | 43.32 | 46 | -32 | 48.79 | 48.4 | -35.2 |
| Ours | **83.79** | **86** | - | **83.08** | **86** | - | **84.05** | **88** | - | **85.48** | **82** | - | **81.79** | **84** | - | **83.64** | **85.2** | - |
| **Style-conditional generation** | | | | | | | | | | | | | | | | | | |
| SRNet | 67.56 | 54 | -32 | 71.51 | 64 | -48 | 52.91 | **84** | -36 | 62.29 | 60 | -24 | 65.94 | 64 | -12 | 64.04 | 65.2 | -30.4 |
| Mostel | **72.54** | 74 | -16 | **76.21** | 80 | -44 | 65.72 | 80 | -40 | 67.89 | **72** | -20 | 68.23 | 76 | 8 | 70.12 | 75.2 | -22.4 |
| SD | 2.42 | 4 | -28 | 4.13 | 6 | 16 | 9.69 | 8 | -12 | 7.91 | 4 | -16 | 3.81 | 2 | -20 | 5.59 | 4.8 | -12.0 |
| SD2 | 4.82 | 6 | -8 | 7.64 | 10 | -8 | 18.13 | 12 | -12 | 12.52 | 6 | 4 | 9.28 | 6 | -4 | 10.48 | 8.0 | -5.6 |
| SD-FT | 23.52 | 16 | -52 | 26.22 | 18 | -20 | 47.16 | 42 | -28 | 43.38 | 40 | -36 | 22.77 | 18 | -34 | 32.61 | 26.8 | -34.0 |
| SD2-FT | 28.56 | 32 | -56 | 34.87 | 24 | -36 | 55.84 | 60 | -40 | 54.79 | 50 | -48 | 36.94 | 48 | -36 | 42.20 | 42.8 | -43.2 |
| Ours | 72.39 | **82** | - | 74.82 | 80 | - | **73.38** | **84** | - | **73.67** | **72** | - | **79.26** | **84** | - | **74.70** | **80.4** | - |

Table 1: Quantitative evaluation on five datasets with style-free generation (top) and style-conditional generation (bottom). Text correctness rate is evaluated by both automatic OCR model (*OCR*) and human labelling (*Cor*). The image naturalness (*Nat*) is evaluated by human, with the difference of vote percentage between each baseline and our method reported. A positive *Nat* value indicates the corresponding baseline is better than ours. All the numbers are in percentage.

**Style-conditional generation** We present the quantitative results of style-conditional generation in Tables 1 and 6. Our key observations are as follows. First, similar to style-free setting, our method achieves the best average text correctness and image naturalness, with 6.5% and 6.9% improvement in *OCR* accuracy and *Cor*, and 5.6% more votes in *Nat*. Second, DiffSte achieves the best font and color correctness compared with other diffusion baselines over all datasets. Different from the diffusion models that take text instructions for specifying the style, the GAN-based methods like Mostel and SRNet take a reference image of rendered texts with the specified font and color. This unfair advantage explains why the GAN-based methods achieve better average color correctness. However, our method DiffSte is still better on font correctness over four of the five datasets with 13.2% more vote percentage in average.

We present qualitative results in Figure 5 and Appendix D.2. It is shown that our model consistently generates correct texts that conform to specified text styles. Although GAN-based methods also generate texts following the specified color, they suffer from poor font correctness. An example could be seen for "survey" text in `ArT` dataset. Besides, GAN-based methods cannot capture the orientation of surrounding texts.

With oblique surrounding texts in `TextOCR` dataset over "better" target text, SRNet and Mostel still tend to generate horizontal texts, resulting in unnaturalness. On the other hand, pretrained diffusion models SD and SD2 completely ignore the styles specified in the instruction, and the problem is mitigated in SD-FT and SD2-FT after fine-tuning, demonstrating that our instruction tuning framework could improve the style controllability. Despite the improvement in style correctness, the fine-

| Method | Synthetic | | ArT | | COCOText | | TextOCR | | ICDAR13 | | Average | |
|---|---|---|---|---|---|---|---|---|---|---|---|---|
| | Font↑ | Color↑ | Font↑ | Color↑ | Font↑ | Color↑ | Font↑ | Color↑ | Font↑ | Color↑ | Font↑ | Color↑ |
| SRNet | -8 | 0 | -16 | 6 | -10 | 8 | 4 | 28 | -36 | 20 | -13.2 | 12.4 |
| Mostel | 0 | 8 | -18 | -18 | -28 | -8 | -24 | 14 | -46 | 14 | -23.2 | 2.0 |
| SD | -86 | -72 | -82 | -72 | -76 | -54 | -82 | -54 | -80 | -76 | -81.2 | -65.6 |
| SD2 | -86 | -70 | -84 | -68 | -62 | -48 | -76 | -64 | -70 | -68 | -75.6 | -63.6 |
| SD-FT | -74 | -36 | -76 | -44 | -66 | -34 | -70 | -38 | -60 | -38 | -69.2 | -38.0 |
| SD2-FT | -72 | -16 | -78 | -28 | -74 | -34 | -68 | -24 | -60 | -32 | -70.4 | -26.8 |

Figure 6: Human evaluation results of font and color correctness for style-conditional generation. We report the difference of votes in percentage between each baseline and our method. A positive value indicates the method is better than ours.

tuned diffusion models still cannot fully capture the specified style compared to our method DiffSte and struggle with incorrect text spelling, demonstrating the effectiveness of our dual-encoder model architecture. For example, SD2-FT generates "JPG" in `Art` dataset when the target text is lowercase "jpg". We present additional experiment results with an ablation study of the proposed character encoder, instruction tuning, and data mixture in Appendix C and quantitative comparison with other diffusion models such as TextDiffuser and SD-XL in Appendix E.

Figure 5: Examples of style-conditional text editing results. The RENDERED row shows the target text and desired style.

### 4.3 Zero-shot Style Combination

In this section, we show that our method can create new text styles by composing basic font attributes in a zero-shot manner benefitting from instruction tuning. We consider two settings: font extrapolation and font interpolation.

**Font extrapolation** In the font extrapolation experiment, we require the model to generate texts in a certain font style but with an unseen format such as *italic, bold*, or both of them. None of these formats were seen during training for the given font. We append format keyword *italic, bold* to those fonts with

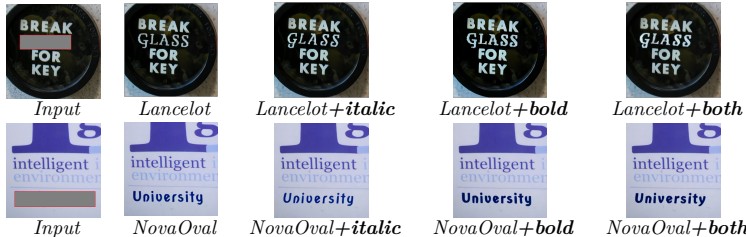

Figure 7: Examples of font extrapolation with our method DIFF-STE. The last three columns are generated with the text instruction template *Write "TEXT" in font FONT-FORMAT*, where the FORMAT is *Bold, Italic* and *Bold Italic*, respectively.

no such format in training data and give the model the extrapolated font name. As shown in Figure 7, the generated texts in the image appear inclined or bolded when we append the format keyword, showing that our method can successfully generalize to unseen font variations while maintaining correctness and naturalness.

**Font interpolation** In the font interpolation experiment, we require the model to generate texts with a mix of two certain font styles, which does not correspond to a particular font style in the real world and was not seen during training. We feed the model with the following instruction template: *Write "TEXT" in font FONT1 and FONT2*, where the FONT1 and FONT2 are the names of two fonts seen in model training. The qualitative results can be seen in Figure 2, where a new font is generated by mixing *MoonDance* and *PressStart2P*. More results are in Appendix D.3.

### 4.4 Editing with Natural Language Instruction

Although we only use static text instruction templates during instruction tuning, we show that our method still shows good scene text editing ability with natural language instructions. The qualitative results can be seen in Figure 8, where we feed the model with instructions such as *"The word "HOTEL" in the color of sky"* instead of the fixed templated instruction seen in training. We show that the model has learned the mapping from instructions to visual text appearances in the generated image following the specified text style. This demonstrates that our instruction tuning framework brings the model the generalization ability to unseen text instructions, which is consistent with previous findings in NLP area Sanh et al. (2021); Wei et al. (2021). The compatibility with natural language instructions makes our model easier to use for non-expert users and enables our model to work in vast downstream applications like chat robots. More results could be seen in Appendix D.4.

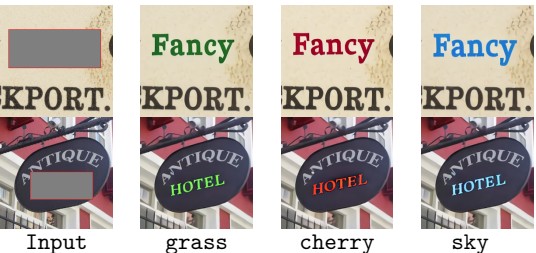

Figure 8: Control text color using natural language instructions with our method DIFFSTE. The color is described with words *"grass colored"*, *"add cherry color to"* and *"in the color of sky"* for the right three columns, respectively.

## 5 Conclusion

In this paper, we present a novel method DIFFSTE to improve pre-trained diffusion models for scene text editing with a dual encoder design that overcomes the limitations of existing diffusion models in terms of text correctness and style control. Our proposed method consists of an instruction encoder for better style control and a character encoder for improved text legibility. Empirical evaluations on five datasets demonstrate the superior performance of DIFFSTE in terms of text correctness, image naturalness, and style controllability. DIFFSTE also shows zero-shot generalization ability to unseen text styles and natural language instructions.

## 6 Broader Impacts

In this paper, we propose a novel method DIFFSTE to adapt pre-trained diffusion models for scene text editing. Our proposed method consists of an instruction encoder for better style control and a character encoder for improved text legibility. DIFFSTE improves text correctness for the pre-trained diffusion model and further allows users to control the generated text styles. However, we also admit that our method can be used to forge signatures or spread misinformation, which might have a significant impact on society. In practice, our method should be appropriately used with careful checks on potential risks. For example, we can adopt recently proposed diffusion-watermarking methods Zhao et al. (2023); Liu et al. (2023); Wen et al. (2023) to identify whether certain text images are generated by our model to identify forged signatures.

## 7 Acknowledgement

The work of Jiabao Ji, Bairu Hou, and Shiyu Chang was partially supported by the Adobe Research Award and the Cisco Research Award.

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

# A    Human Evaluation Details

In this section, we provide a summary of the human evaluations conducted for both style-free and style-conditional generation settings. As outlined in Section 3, we perform human evaluations on Amazon Mturk[3] for both style-free and style-conditional generation settings. In total, we design six human evaluation tasks to assess the performance of our method from three aspects: ① the accuracy of the generated text in the image; ② the naturalness of the generated image; ③ the correctness of text style in terms of font and color.

**Human evaluation for style-free generation**    To evaluate style-free image generation, we assess the generated images from two aspects: ① *Cor*, text correctness, where evaluators check whether the generated image matches the exact target text. The user interface is shown in Figure 9. ② *Nat*, image naturalness, where evaluators select the more natural and visually coherent image between those generated by our method and another baseline method. In other words, the chosen image should look more like a natural image, and the completed missing part should be visually coherent with the surrounding region. The user interface is shown in Figure 10.

**Human evaluation for style-conditional generation**    To evaluate style-conditional generation, we assess the generated images from three aspects: ① *Cor*, text correctness, which is the same as the style-free generation setting. ② *Nat*, image naturalness, where evaluators select a more natural image between those generated by our method and other baseline methods. We note that in the style-conditional generation setting, the specified color in instructions may not be coherent with the surrounding region. Therefore, we exclude the coherence criterion for image naturalness in this evaluation. The user interface for is depicted in Figure 11. ③ *Font* and *Color*, style correctness, where evaluators compare images generated by our method and other baseline methods and select a better one based on how similar the generated texts are to a given reference image in terms of font or color, respectively. The user interfaces are shown in Figure 12 and Figure 13.

---

**Instructions:**

---

Please read the instructions carefully. Failure to follow the instructions will lead to rejection of your results.

In this task, you will see an image in which a certain area is highlighted with a red box. Your task is to determine whether the text's spelling in this area matches the target text shown above the image. It is important that the spelling need to be exactly the same as the target text. Specifically, the following cases are considered incorrect:

- Additional characters, e.g. the generated text is 'apples' but the target text is 'apple'.
- Missing characters, e.g. the generated text is 'appe' but the target text is 'apple'.
- Mismatched capitalization, e.g. the generated text is 'Apple' but the target text is 'apple'.

We provide an example to help you understand the criteria.

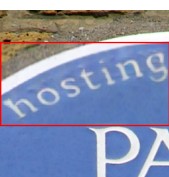

In this example, you should choose Yes. It is important that all we care about is the correctness of text spelling within the highlighted area. Although the text in this region is blurry, it has the correct target text 'hosting'.

---

Figure 9: Instructions of human evaluation on text correctness.

[3]https://www.mturk.com

**Instructions:**

Please read the instructions carefully. Failure to follow the instructions will lead to the rejection of your results. In this task, you will be asked to judge and compare the quality of two AI-edited images. Specifically, you will first see one source image with some parts missing. Then, you will see two candidate images, where the missing part is completed by two AI algorithms. Your task is to determine which of the two candidate images is more visually natural and coherent. It is important that we only care about the visual naturalness and coherence of the completed missing part regardless of anything else, such as the correctness of the text spelling.

There are two criteria for this task:

• First, the edited image should look like a natural image. It should not contain a lot of artifacts, distortion, or non-commonsensical scenes. The completed part should not be blurry and there should be no clear boundary between this completed part and the surrounding region.

• Second, the completed missing part should be visually coherent with the surrounding region. If there is any other text in the surroundings, the text in the completed region should match the style of these texts in terms of shape and color.

We provide an example to help you understand the criteria.

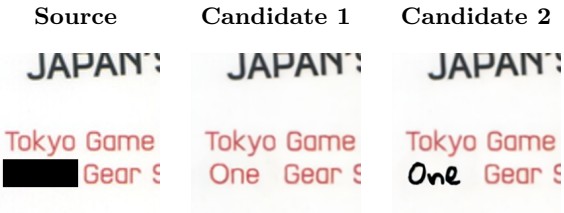

In this example, you should choose Candidate 1 as the better one. It is important that the completed text in the missing region matches the style of surrounding texts. Candidate 2, however, has a black text, which does not align well with other red surrounding texts, which does not meet our criteria on coherence.

Figure 10: Instructions of human evaluation based on image naturalness and coherence.

**Instructions:**

Please read the instructions carefully. Failure to follow the instructions will lead to rejection of your results. In this task, you will be asked to judge and compare the quality of two AI-edited images. Specifically, you will first see one source image with some parts missing. Then, you will see two candidate images, where the missing part is completed by two AI algorithms. Your task is to determine which of the two candidate images is more visually natural. It is important that we only care about the visual naturalness of the completed missing part regardless of anything else, such as the correctness of the text spelling. There are one criteria for this task:

• The edited image should look like a natural image. It should not contain a lot of artifacts, distortion, or non-commonsensical scenes. The completed part should not be blurry and there should be no clear boundary between this completed part and the surrounding region.

We provide an example to help you understand the criteria.

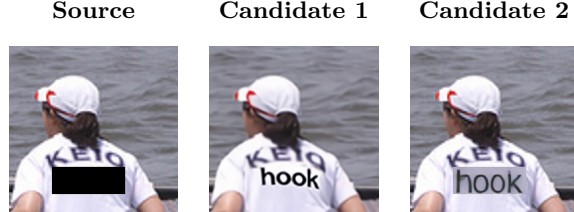

In this example, you should choose Candidate 1 as the better one. It is important that the completed part of the image is not blurry and there is no clear boundary between completed part and surrounding image. Candiate 2, however, has a clear boundary between the completed part and the surrounding region, which does not meet our creteria on naturalness.

Figure 11: Instructions of human evaluation on image naturalness.

**Instructions:**

Please read the instructions carefully. Failure to follow the instructions will lead to rejection of your results. In this task, you will be asked to judge and compare the quality of two AI-edited images. Specifically, you will first see a reference image with a certain text that specifies the desired font style we want the AI model to generate. Then, you will see two candidate images, both of which have some parts highlighted with a red box. Your task is to determine which of the two candidate images appears more similar to the reference image in terms of font style. More specifically, font style refers to the shape of characters and whether the text is bold or italic. It is important to note that we only care about the font style of the text, regardless of anything else, such as the correctness of the text spelling or the color of the text. There is a "hard-to-tell" option available if you cannot decide which is better, but please try your best to avoid using this option.
We provide an example to help you understand the criteria.

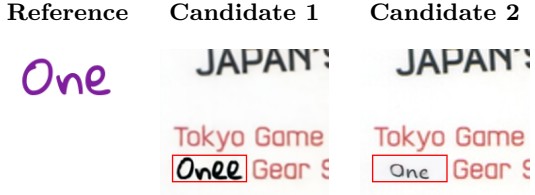

In this example, you should choose Candidate 1 as the better one. It is important that all we care about is whether the font style of text is simiar to the reference image. You can see that the shape of characters in Candidate2 is not the same as the reference image, e.g. the chracter O is not similar. Therefore, you should choose Candidate1 as the better one despite that the text spelling in Candidate 1 is not the same as the reference image.

Figure 12: Instructions of human evaluation on font correctness.

**Instructions:**

Please read the instructions carefully. Failure to follow the instructions will lead to rejection of your results. In this task, you will be asked to judge and compare the quality of two AI-edited images. Specifically, you will first see a reference image with a certain text that specifies the color of the text that we want the AI model to generate. Then, you will see two candidate images, both of which have some parts highlighted with a red box. Your task is to determine which of the two candidate images appears more similar to the reference image in terms of the text color. It is important to note that we only care about the color of the text, regardless of the spelling correctness or font type. There is a "hard-to-tell" option for you to choose if you cannot decide which is better, but please try your best to avoid using this option.

We provide an example to help you understand the criteria.

In this example, you should choose Candidate 2 as the better one. It is important that all we care about is whether the color of text is simiar to the reference image. Candidate 2 is better since the text in the highlighted area looks more purple, which is similar to the reference image, despite that the text content is not the same.

Figure 13: Instructions of human evaluation on color correctness.

# B  Implementation Details

## B.1  Dataset Details

As mentioned in Section 4.1, we construct the synthetic dataset using a publicly available engine Synthtiger Yim et al. (2021). We randomly selected 100 font families in the google fonts library and 954 named XKCD colors to render the texts. During the creation process, we first randomly sample several English words or numerical numbers (1-5) and render them using a random color and font style. Then we apply the same rotation and perspective transformation to these rendered scene texts and place them on a $256 \times 256$ background image non-overlappingly. In training time, a random word in the image is masked out and the model is trained to restore the full image including the text in the masked part. Figure 14 shows examples of the created synthetic data and the masked input for model training. Besides the synthetic data, we also include real-world OCR datasets in the training set, where the word in image is masked out for training. In total, we collect 1.3M training examples with 600K synthetic examples and 700K real-world examples.

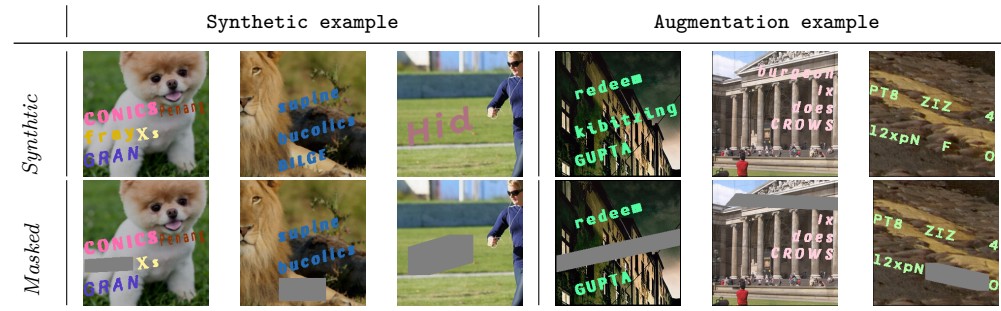

Figure 14: Example images of synthetic data(first row) and the masked input(second row) we use to fine-tune the model.

## B.2  Model Details

**GAN-based methods**   We implement the GAN-based method, SRNET and MOSTEL using their publicly released code. We created additional synthetic examples where different texts of the same style are put at the same location in the background image to construct paired images for style-transfer training, which is only used in fine-tuning SRNET. Real-world datasets are not included in training SRNET since it requires paired images with same background but different texts, which is not given in real-world datasets. For SRNET, we follow the hyperparameters reported in their paper to fine-tune the released model. For MOSTEL, fine-tuning the released model on the created synthetic data does not improve its performance on real-world datasets. Therefore the released checkpoint is used for comparison.

**Diffusion-based methods**   All diffusion-based methods are built upon *diffusers*[4]. For the pre-trained diffusion models, we use stable-diffusion-inpainting[5](SD) and stable-diffusion-2-inpainting[6](SD2). We further fine-tune SD and SD2 using our instruction tuning framework, where the resulting models are termed as SD-FT and SD2-FT. For our method DIFFSTE, we use SD as the backbone[7] and randomly initialize the character-embedding layer and corresponding cross-attention weights. The dimension of character embedding is 32, and the number of cross-attention heads is 8. We use the same hyperparameters to train SD-FT, SD2-FT, and our method DIFFSTE. The batch size is set to 256. We use the AdamW optimizer with a fixed learning rate $5e-5$ to train the model for 15 epochs. In total, the training has 80k steps, which requires approximately two days of training time using eight Nvidia-V100 gpus.

**Data augmentation**   In our initial experiment, we observed a strong correlation between the mask shape and the visual layout of text in the generated image. Specifically, the model tends to duplicate characters to fill the entire masked region even if there are only a few characters to generate. To mitigate this bias, we augment the masks to be larger than the text in the original image. Specifically, we extend the mask region

---

[4]https://huggingface.co/docs/diffusers

[5]https://huggingface.co/runwayml/stable-diffusion-inpainting

[6]stabilityai/stable-diffusion-2-inpainting

[7]SD2 is not released by the time of our implementation.

along the direction of the masked word while keeping sure that no other texts are included in the mask. Examples can be seen on the right of Figure 14.

## C  Ablation Study

Our method DIFFSTE improves upon the pre-trained SD model from two aspects: ① the dual-encoder design with the newly added character encoder; ② the instruction tuning framework that helps model understand the task of scene text editing. In this section, we study the effectiveness of these two designs. To obtain a deeper understanding of the effectiveness of synthetic and real data, we also conduct an ablation study on the data mixture ratio.

**Character encoder**  We evaluate the effectiveness of the character encoder by comparing the pre-traiend backbone model SD, instruction-tuned SD without character encoder, and our method DIFFSTE, the instruction-tuned SD with character encoder. Table 2 shows the quantitative evaluation results on five datasets, where these three models are termed as SD, +INST, and +INST+CHAR respectively. Although instruction tuning (+INST) improves the ability of the pre-trained model SD to generate correct texts with an average of 30% increase in OCR accuracy, the text correctness is way lower than the +INST+CHAR, which is additionally trained with the character encoder. This demonstrates that the dual-encoder design could effectively improve the model's ability to capture the spelling of target text.

| Method | Synthetic | | ArT | | COCOText | | TextOCR | | ICDAR13 | | Average | |
|---|---|---|---|---|---|---|---|---|---|---|---|---|
| | $OCR\uparrow$ | $Cor\uparrow$ | $OCR\uparrow$ | $Cor\uparrow$ | $OCR\uparrow$ | $Cor\uparrow$ | $OCR\uparrow$ | $Cor\uparrow$ | $OCR\uparrow$ | $Cor\uparrow$ | $OCR\uparrow$ | $Cor\uparrow$ |
| **Style-free generation** | | | | | | | | | | | | |
| SD | 3.12 | 6 | 5.39 | 6 | 12.46 | 10 | 8.22 | 4 | 4.56 | 6 | 6.75 | 6.4 |
| +INST | 29.51 | 34 | 32.08 | 34 | 51.62 | 60 | 48.91 | 44 | 25.29 | 30 | 37.48 | 40.4 |
| +INST+CHAR | **83.79** | **86** | **83.08** | **86** | **84.05** | **88** | **85.48** | **82** | **81.79** | **84** | **83.64** | **85.2** |
| **Style-conditional generation** | | | | | | | | | | | | |
| SD | 3.12 | 6 | 5.39 | 6 | 12.46 | 10 | 8.22 | 4 | 4.56 | 6 | 6.75 | 6.4 |
| +INST | 23.52 | 16 | 26.22 | 18 | 47.16 | 42 | 43.38 | 40 | 22.77 | 18 | 32.61 | 26.8 |
| +INST+CHAR | **72.39** | **82** | **74.82** | **80** | **73.38** | **84** | **73.67** | **72** | **79.26** | **84** | **74.70** | **80.4** |

Table 2: Quantitative evaluation on five datasets with style-free generation (top) and style-conditional generation (bottom) of the pre-trained backbone model SD, the fine-tuned SD without character encoder (+INST) and the fine-tuned model with character encoder, *i.e.* DIFFSTE(+INST+CHAR). All the numbers are in percentage.

**Instruction tuning**  We evaluate the effectiveness of the instruction tuning framework by comparing DIFFSTE by comparing DIFFSTE with three other models that share the same backbone and character encoder but are trained using different instructions. The models evaluated are trained using instructions without font information, color information, and neither of them, respectively. We denote them as -FONT, -COLOR, and -BOTH.

| Method | Synthetic | | ArT | | COCOText | | TextOCR | | ICDAR13 | | Average | |
|---|---|---|---|---|---|---|---|---|---|---|---|---|
| | $Font\uparrow$ | $Color\uparrow$ | $Font\uparrow$ | $Color\uparrow$ | $Font\uparrow$ | $Color\uparrow$ | $Font\uparrow$ | $Color\uparrow$ | $Font\uparrow$ | $Color\uparrow$ | $Font\uparrow$ | $Color\uparrow$ |
| DIFFSTE | - | - | - | - | - | - | - | - | - | - | - | - |
| -FONT | -72 | - | -64 | - | -76 | - | -70 | - | -68 | - | -70.0 | - |
| -COLOR | - | -54 | - | -66 | - | -72 | - | -84 | - | -68 | - | -68.8 |
| -BOTH | -70 | -58 | -68 | -74 | -80 | -78 | -78 | -80 | -78 | -64 | -74.8 | -70.8 |

Table 3: Human evaluation results of font and color correctness for no-instruction/partial-instruction tuning. The difference of votes in percentage between the baseline method and our method is reported. A positive value indicates the method is better than ours.

To evaluate their ability in understanding style specifications in the instructions, we compare the three models with DIFFSTE under the style-conditional generation setting. Similar to Section 4.2, we ask human evaluators to make a pairwise comparison between images generated by these models and DIFFSTE, which is

trained on full instructions with font and color information on 50 randomly picked images for each dataset. The quantitative results are shown in Table 3. We observe that removing certain style information in the instructions would lead to a decrease in corresponding style correctness dramatically. For example, without font instructions, -FONT receives 70.0% fewer votes in terms of font correctness, while a 68.8% average drop is observed for color correctness for -COLOR. Without both font and color, -BOTH, descend 74.8% and 70.8% in terms of font and color correctness, even more than -FONT and -COLOR. This further demonstrates the effectiveness of our instruction tuning framework in teaching the model to align text instruction with visual appearances.

**Synthetic v.s. Real data mixture ratio** To comprehensively study the mixture ratio between synthetic and real-world data, we train two additional versions of our method. Specifically, we train one model on synthetic data only and another on real-world text images only. We compare these two trained models to the original DIFFSTE in both style-free and style-conditional settings, as described in the main paper. The OCR results are presented in Table 4, and the generation examples can be found in Figure 15 and Figure 16. We highlight the following observations:

First, all three models achieve significantly better text correctness compared to the vanilla stable-diffusion 1/2 model and even the fine-tuned SD1/2. This demonstrates that our dual encoder design effectively helps the base diffusion model understand correct word spelling and generate accurate visual texts.

Second, training on real or synthetic data only hurts the text's correctness. The combined DiffSTE model achieves more than 20% absolute percentage improvement.

Third, training on synthetic images only hurts the image quality. As can be seen in the fourth row of Figure 15, the visual texts generated by the model trained on synthetic data only do not align well with other surrounding texts. This issue arises from the limited text layouts and styles in the synthetic dataset we use. Therefore, we include real data in DIFFSTE training to provide the variability of background scenes, enrich the text styles, and enable the model to better generalize to test-time real-world text images.

Finally, We also want to highlight that synthetic data builds the foundation for the editing ability of DIFFSTE. We explicitly construct the data to teach the model to follow editing instructions and infer style from surrounding texts (see Sec 3.2 in the main paper), which is the main focus of our paper. As can be seen in the fourth row in Figure 16, the model trained on real data only does not understand the style control instructions, thus the generated texts are always a plain text style, either matching the surrounding text or the background similar to style-free generation. The text correctness is also affected as the model never sees style specification instructions in training (failed generation in row 4 column 4 of Figure 16).

| Style-free generation | | | | | |
|---|---|---|---|---|---|
| Model | ArT | COCOText | TextOCR | ICDAR13 | Average |
| DiffSTE | 83.08 | 84.05 | 85.48 | 81.70 | 83.64 |
| -REAL | 46.34 | 64.17 | 64.83 | 55.47 | 57.73 |
| -SYN. | 52.31 | 70.31 | 68.27 | 62.11 | 63.25 |
| SD | 5.39 | 12.46 | 8.22 | 4.56 | 7.67 |
| SD2 | 7.28 | 19.34 | 11.88 | 9.13 | 11.91 |
| SD-FT | 32.08 | 51.62 | 48.91 | 25.29 | 39.48 |
| SD2-FT | 46.17 | 56.13 | 60.82 | 43.32 | 51.61 |
| Style-cond generation | | | | | |
| Model | ArT | COCOText | TextOCR | ICDAR13 | Average |
| DiffSTE | 74.82 | 73.38 | 73.67 | 79.26 | 75.33 |
| -REAL | 32.35 | 45.12 | 43.28 | 34.21 | 38.74 |
| -SYN. | 44.21 | 58.93 | 60.42 | 54.32 | 54.47 |
| SD | 4.13 | 9.69 | 7.91 | 3.81 | 6.39 |
| SD2 | 7.64 | 18.13 | 12.52 | 9.28 | 11.89 |
| SD-FT | 26.22 | 47.16 | 43.38 | 22.77 | 34.88 |
| SD2-FT | 34.87 | 55.84 | 54.79 | 36.94 | 45.61 |

Table 4: OCR correctness on real-world datasets with style-free generation (top) and style-conditional generation (bottom) for three versions of DiffSTE. The correctness is evaluated by the automatic OCR model and all numbers are reported in percentage. -REAL and -SYN. represent a model trained on real-world text images only and synthetic images only respectively using our method.

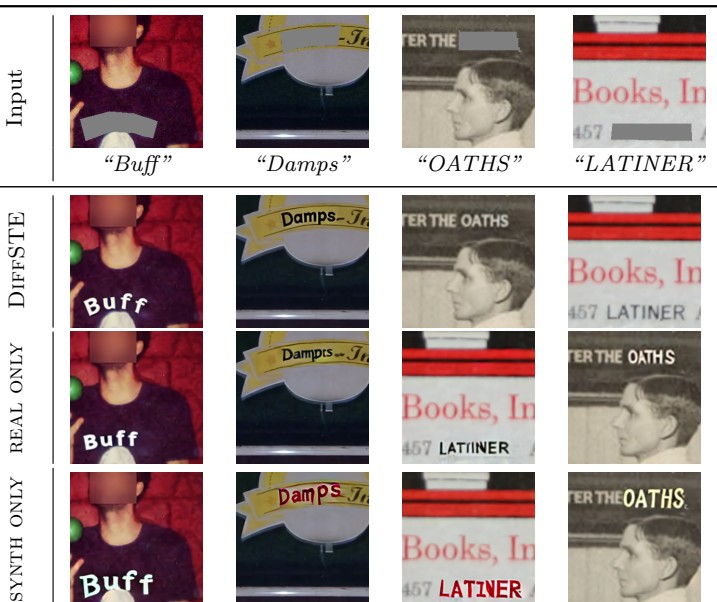

Figure 15: Examples of style-free text editing results with three versions of DiffSTE. REAL ONLY and SYNTH ONLY denotes the model trained on real-world text images and synthetic text images only using our method.

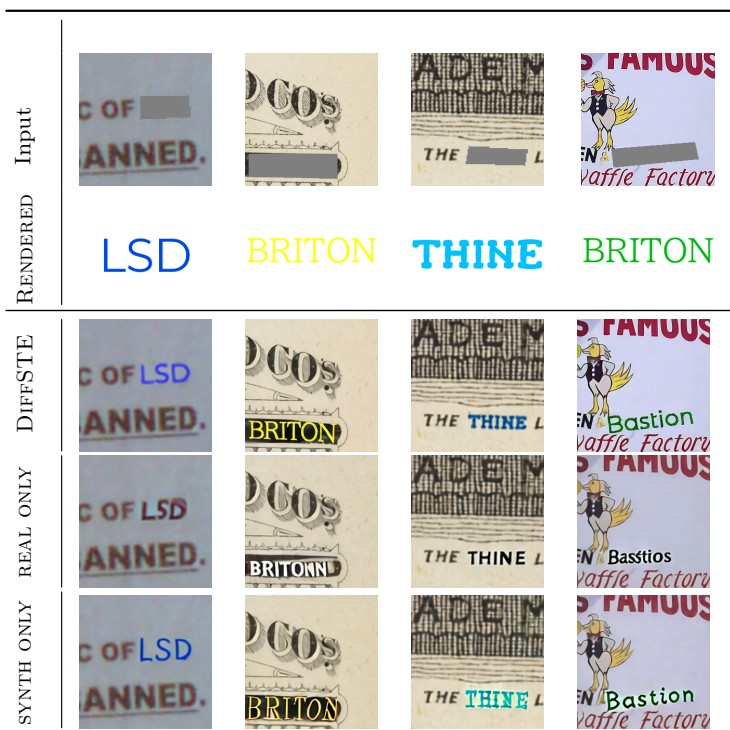

Figure 16: Examples of style-condition text editing results with three versions of DiffSTE. REAL ONLY and SYNTH ONLY denotes the model trained on real-world text images and synthetic text images only using our method.

# D   Additional Qualitative Results

## D.1   Additional Style-free Qualitative Results

We provide additional generated images for editing different texts in the same image by our method DIFFSTE in Figure 17. The first row shows the target text to generate, and each column shows the different generated images on the target text. DIFFSTE consistently generates correct visual text, and the texts naturally follow the same text style, *i.e.* font, and color, with other surrounding texts. The orientation of generated texts also aligns with other texts properly. In addition to the normally shaped masks shown in Figure 17, DIFFSTE is able to generate texts within arbitrarily shaped masks. We provide the qualitative results in Section D.5.

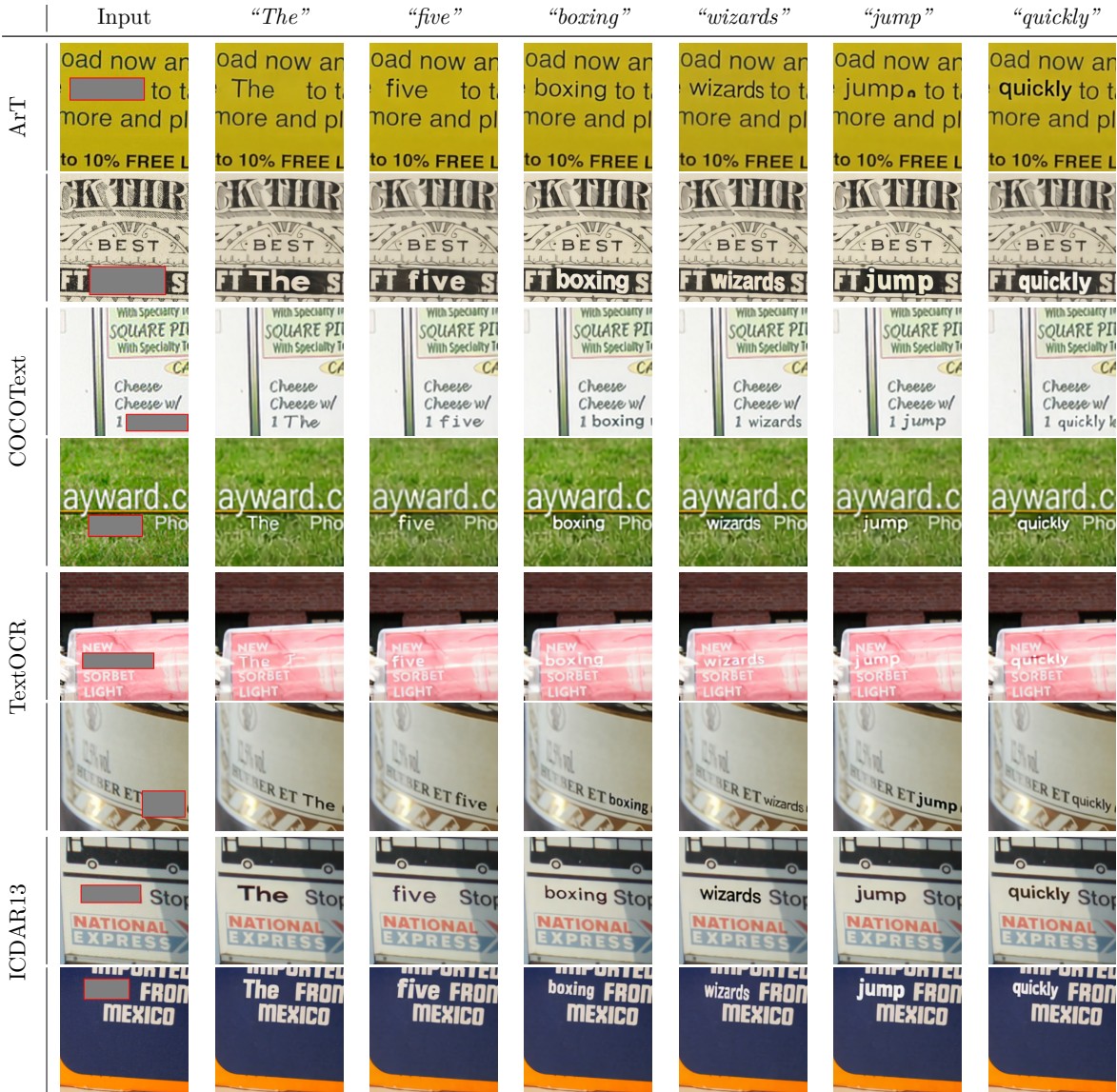

Figure 17: Examples of style-free text editing results on four real-world datasets.

### D.2 Additional Style-conditional Qualitative Results

We provide additional examples of editing different texts with specified styles in the same image by our method DIFFSTE in Figure 18. The first row shows the target visual text style in the generated image. Our method DIFFSTE generate correct texts following the specified text styles consistently. The generated texts naturally fit the surrounding region with suitable size and orientation. For example, texts in the last row in Figure 18 are placed in the same orientation with surrounding texts.

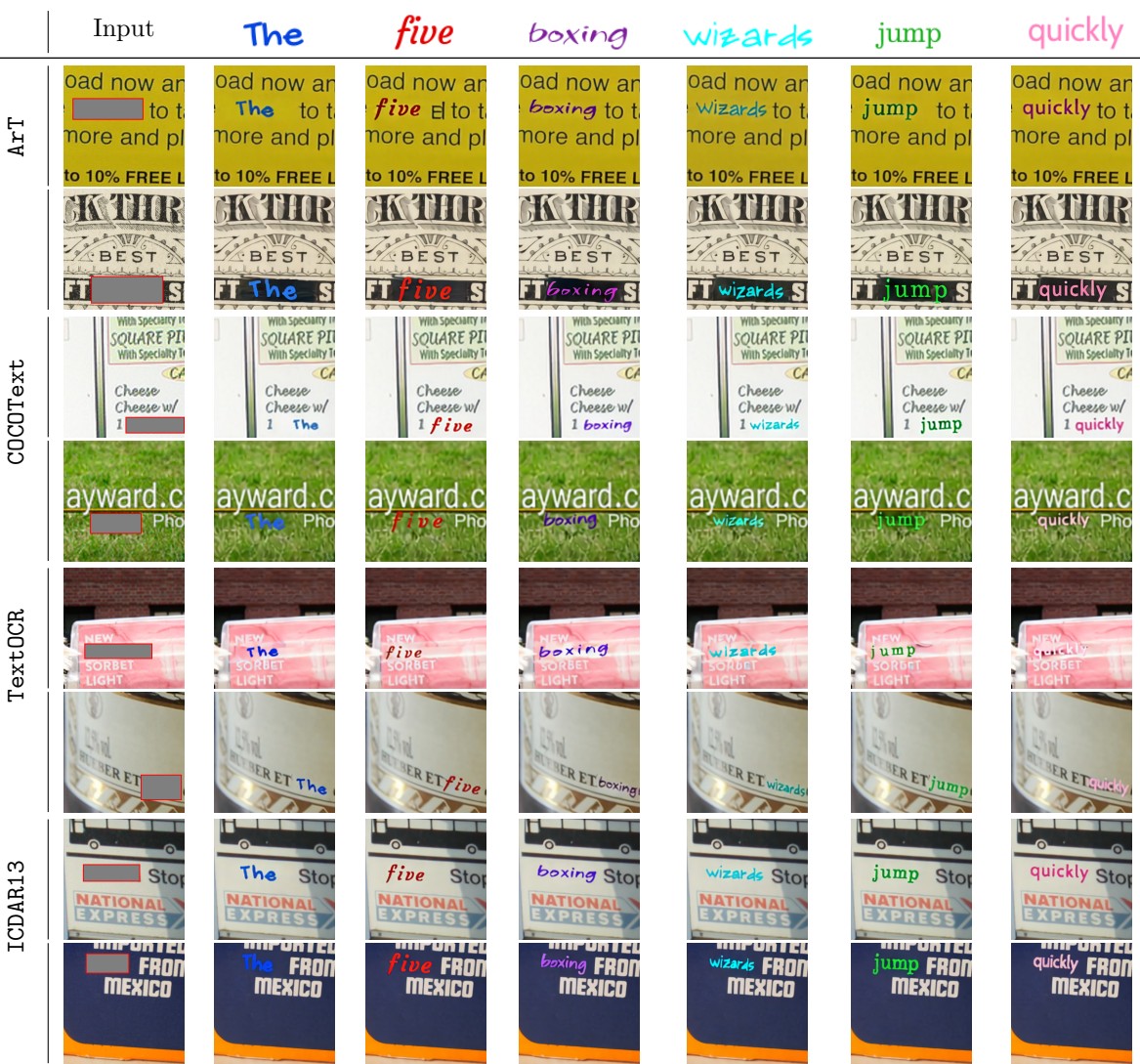

Figure 18: Examples of style-conditional text editing results on four real-world datasets.

### D.3 Additional Qualitative Results with Zero-Shot Style Combination

**Font extropolation** As mentioned in Section 4.2, our method DIFFSTE is able to extend seen font style to unseen formats such as *italic, bold* and both of them. We provide additional examples for font extrapolation in Figure 19. Each row shows the input masked image and generated texts using the specified font and the font with other format combinations. The instruction for font combination is generated in this template *Write "TEXT" in font FONT-FORMAT*. As seen in Figure 19, the generated texts are bolded and inclined following the given format keyword, showing the zero-shot generalizability of DIFFSTE.

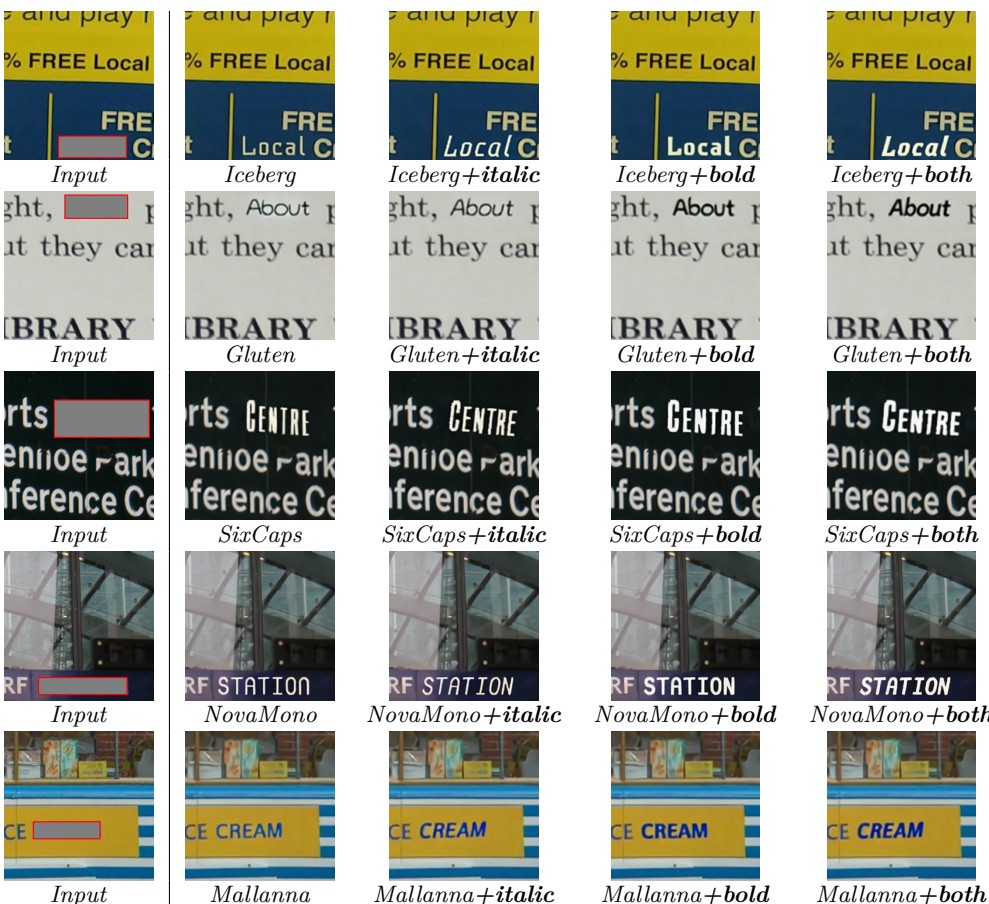

Figure 19: Examples of font extrapolation with our method DIFFSTE.

**Font interpolatoin**   As mentioned in Section 4.2, our method DiffSte is able to create a new font style by mixing two seen font styles. We provide additional examples for font interpolation in Figure 20. To generate instructions for mixing font styles, we use the template *Write Text in font Font1 and Font2*, where Font1 and Font2 are two font names seen in training. As seen in Figure 20, DiffSte is able to mix the two font styles by merging the glyphs of the same character in different fonts using a simple natural language instruction.

Figure 20: Examples of font interpolation with our method DiffSte.

### D.4 Additional Qualitative Results with Natural Language Instructions

We provide additional examples for controlling text color with natural language instructions. In Figure 21, each row shows examples generated using the following instructions generated by ChatGPT[8]: ① *A mango-colored word "Text". ② The word "Text" resembles the color of an eggplant. ③ The word "Text" is colored in a delicate, ladylike shade of lilac. ④ The color of word "Text" is like a blazing inferno. ⑤ The color of word "Text" is deep, oceanic, reminiscent of the deep sea. ⑥ The color of word "Text" used is the fresh, vibrant color of a springtime tree.* As seen in Figure 21, DiffSTE is able to generalize to unseen natural language instructions, which shows the effectiveness of our instruction learning framework in helping the model learn the mapping from natural language instructions to visual text appearances in the generated images.

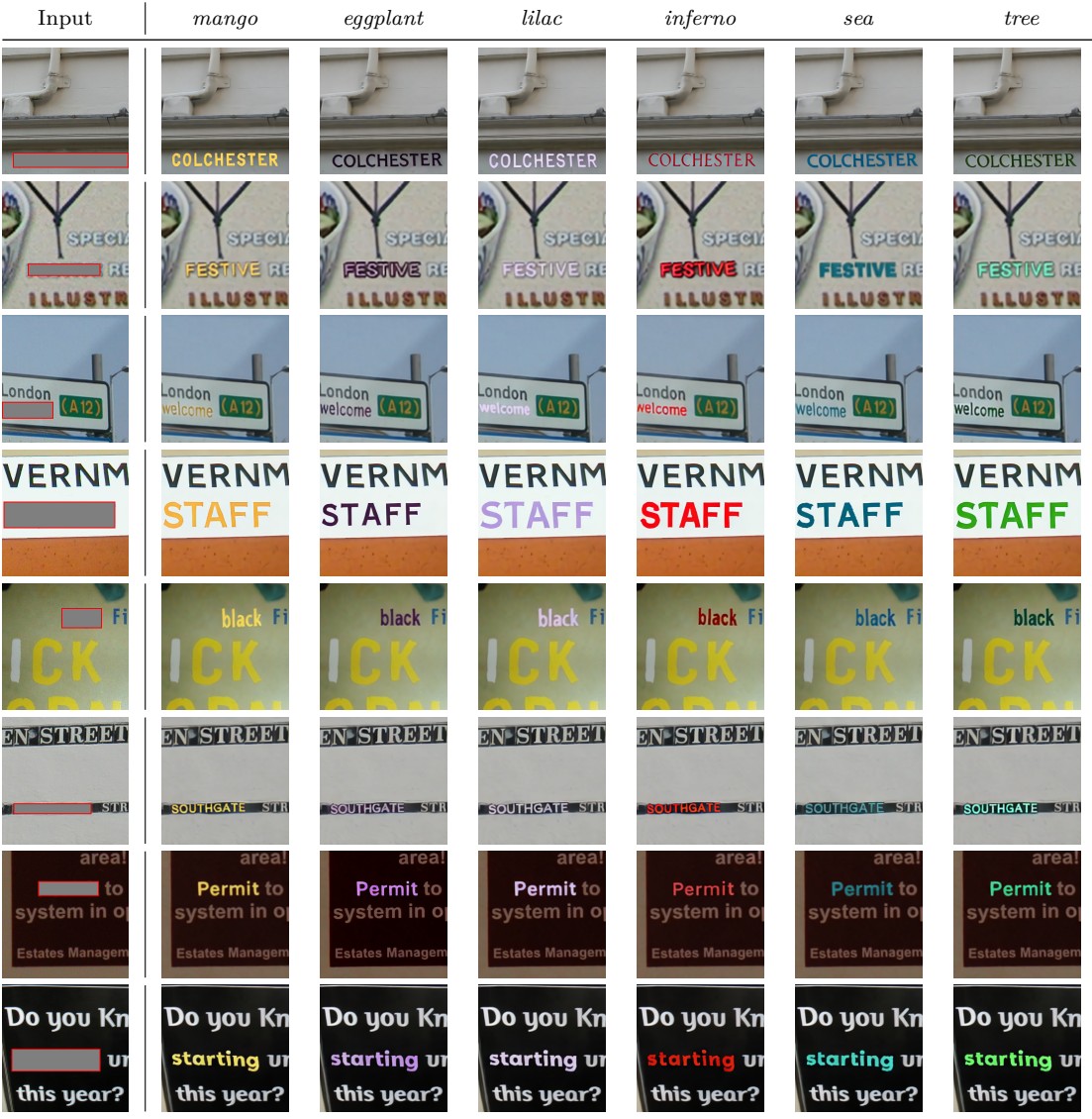

Figure 21: Control text color using different natural language instructions with DiffSTE.

---

[8]We provide this instruction to ChatGPT: *Show me some sentences describing the color of a text similar to this "The text is in the color of grass"* and manually exclude unnecessary details from the generated sentences.

### D.5 Additional Qualitative Results with Arbitrarily-Shaped Mask

We provide additional examples for our method DIFFSTE editing texts in arbitrarily shaped masks in Figure 22. As seen in Figure 22, our method is able to generate visually appealing text layouts within highly curved and lengthy masks, even when the text to generate is short. Notably, in the fourth row of Figure 22, DIFFSTE successfully generates the words *"The"* and *"five"* in an appropriate layout within the curved mask. Each character is placed in a suitable orientation based on the geometry of the given mask region. In other examples, this is also observed for long texts such as *"boxing"*.

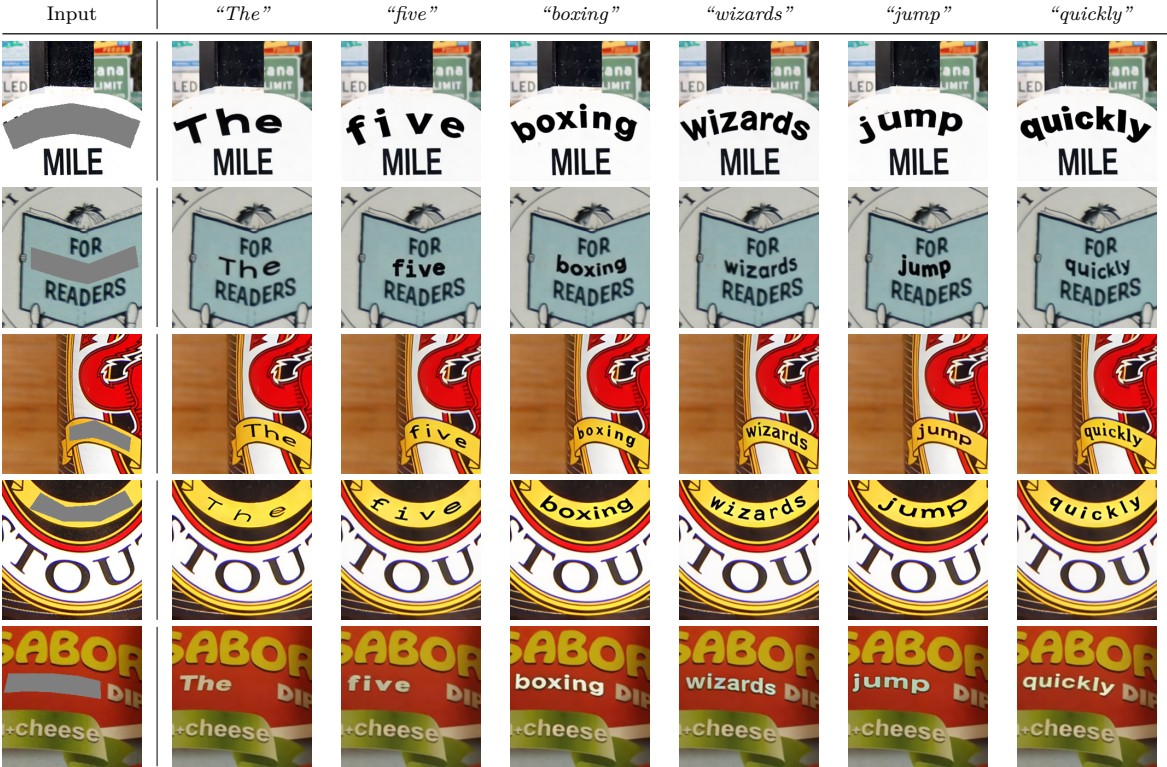

Figure 22: Generate texts within an arbitrarily shaped mask.

# E  More discussions

### E.1  Comparison with other diffusion-based text editing model.

We conduct a comparison with two more diffusion-based text editing models, Text-diffuser Chen et al. (2023b) and SDXL Podell et al. (2023) on our test dataset in both style-free and style-cond settings. The quantitative results are presented in Table 5, and example generations are listed in Figure 23 and Figure 24. We highlight the following observations:

**Significant improvements over SDXL.** Compared to the backbone model we use, Stable-diffusion-v1-5-inpainting (SD) (865M), SDXL has much more than twice the parameters (2.7B). Nevertheless, the proposed dual-encoder design and instruction tuning framework can significantly improve SD's scene text editing capability. Equipped with our method, the SD/SD2 outperforms the much larger model, SDXL, with 70% absolute improvement in text correctness rate.

**Comparable performance as TextD-iffuser with less input informa-tion**. Our method achieves state-of-the-art performance and is comparable to TEXTDIFFUSER. This is impressive, especially considering TEXTDIFFUSER has an advantage in the input. Particularly, TEXTDIFFUSER first renders a layout image that already contains the ground-truth content, font, color, and other information about the scene text to be added or edited. After that, the diffusion model will take the user prompt and the layout image for generation. In comparison, our method only relies on the user prompt but still achieves good performance on text correctness. We additionally discuss the divergence in input information in point 3 below.

| Style-free generation | | | | | |
|---|---|---|---|---|---|
| Model | ArT | COCOText | TextOCR | ICDAR13 | Average |
| DIFFSTE | 83.08 | 84.05 | 85.48 | 81.70 | 83.64 |
| TEXTDIFFUSER | 84.29 | 84.86 | 88.20 | 82.14 | 84.87 |
| SDXL | 5.41 | 9.74 | 8.67 | 6.20 | 7.51 |
| Style-cond generation | | | | | |
| Model | ArT | COCOText | TextOCR | ICDAR13 | Average |
| DIFFSTE | 74.82 | 73.38 | 73.67 | 79.26 | 75.33 |
| TEXTDIFFUSER | 72.77 | 74.34 | 80.67 | 75.43 | 75.80 |
| SDXL | 4.83 | 9.75 | 6.89 | 5.13 | 6.65 |

Table 5: OCR correctness on real-world datasets with style-free generation (top) and style-conditional generation (bottom) for DIFFSTE, TEXTDIFFUSER and SDXL. The correctness is evaluated by the automatic OCR model, and all numbers are reported in percentages.

**Divergence between TextDiffuser and our method.** In our paper, we mainly consider more flexible scene text editing, where users only need to specify the style information (e.g., color and font) and the content of texts through prompts. The diffusion model can also infer a style that is coherent with the background and other parts of the image. In comparison, TEXTDIFFUSER focuses on specifying most text information via the layout image. While achieving better control of the generation, flexibility can sometimes be hurt. For example, we list two scenarios where the layout image in TEXTDIFFUSER cannot accurately and naturally specify the target style as follows:

- **Scene text editing on curved regions or perspective changes:** In the first and second column in Figure 23 and Figure 24, we illustrate examples where the target region is highly curved or has perspective changes. TEXTDIFFUSER cannot provide high-quality and visually appealing text layouts within such curved regions. This is mainly because TEXTDIFFUSER highly depends on the rendered layout image. Once the layout image for target scene texts cannot be generated perfectly (which is exactly the case in those examples), the generation quality would be largely decreased. In comparison, our method provides a more flexible and robust scene text editing for such scenarios and can generate significantly better images.

- **TextDiffuser cannot support style editing.** As can be seen in the fourth row of Figure 24, TEXTDIFFUSER cannot understand the style specification instructions and simply conducts style-free generation by generating texts following the color of other surrounding texts (columns 1, 3, 4). In comparison, our instruction-tuned DifSTE model generates text faithful to the specified text style as shown in the third row in Figure 24, which offers users more flexibility to edit the text style.

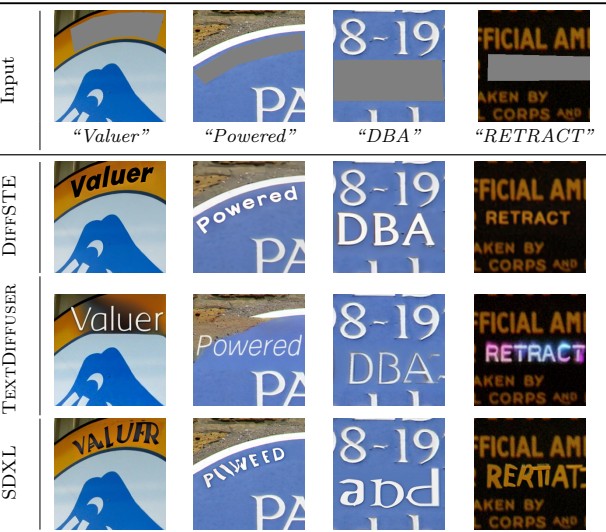

Figure 23: Examples of style-free text editing results with DiffSTE, TextDiffuser and SDXL.

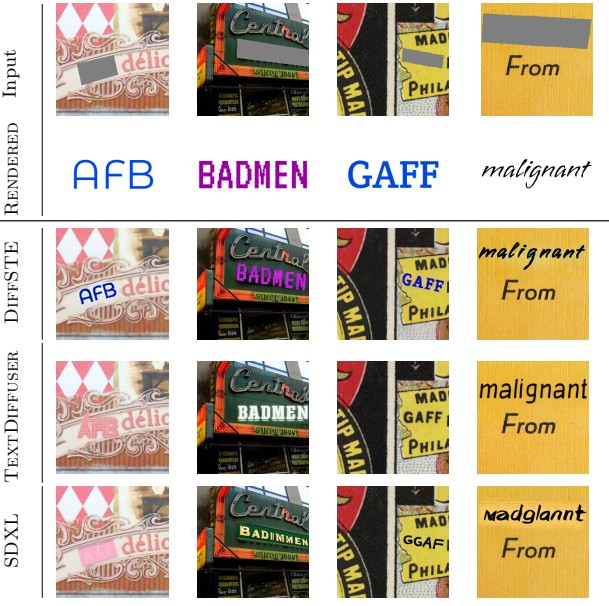

Figure 24: Examples of style-conditional text editing results with DiffSTE, TextDiffuser and SDXL.

## E.2 Limitation and Future work

Although DiffSTE generally produces natural scene texts, as evidenced by the human evaluations detailed in the main paper. Sometimes, the generation results may not be satisfiable. We attribute the unnaturalness to three main reasons: ① lack of surrounding scene texts, which makes inferring the correct text style very challenging. It's evident that when more surrounding texts are present, DiffSTE is better able to generate natural-looking texts; see examples in columns 1 and 6 of Figure 4. ② low resolution ($256 \times 256$) training image in our dataset, which is limited by our computational resources. ③ the synthetic data included in our dataset, where rendered text is simply placed onto a background image. Enhancing the quality of our training image dataset is a potential avenue for improving overall image quality. We regard this as a key area for future work.

