# OpenReview forum: "Improving Diffusion Models for Scene Text Editing with Dual Encoders"
_TMLR — Accepted by TMLR_

### Review · Reviewer_ZirC · 2023-09-23

**Summary Of Contributions:**

This work propose a model DiffSte built upon pre-trained SD model together with an instruct tuning method. A dual encoder architecture is designed  for better text legibility and style control. Instruction tuning framework is enable font style control and blending. The generated results show the superiority of the proposed method over other baselines.

**Audience:**

Yes

**Broader Impact Concerns:**

No broader impact concerns are required.

**Claims And Evidence:**

Yes

**Requested Changes:**

Comparisons with diffusion-based methods such as Text-diffuser  (code & model released)  or  DiffUTE (if released), or text2image method such as SDXL  (code & model released)  can be presented.

**Strengths And Weaknesses:**

Strengths:

- Dual-encoder architecture is design for input conditions of text character and font style. The char encoder extract the text shape and spatial information and fuse such information though cross-attention. The inst encoder extract the style information given the instruction.

- Instruction tuning technology is applied to train these modules to equip the model with the ability of font stylization.

- Through style instruction, the trained model is able to do style blending.

Weakness:

- Comparisons with diffusion-based related works are necessary, such as Text-diffuser  (code released) and  DiffUTE (if released) or text2image method such as SDXL  (code & model released). Although these methods are not explicitly conditioned on input font style condition, they capture font styles from the input image or input prompt (e.g. SDXL).

---

> ### Author Response · Authors · 2023-12-22
> **Response to Reviewer ZirC**
>
> We thank Reviewer ZirC for the insightful suggestions for our paper. In the reminder, we want to address the key points raised in reviews.
>
> **Q1: Comparison with diffusion-based text-diffuser and SDXL.**
>
> We appreciate the Reviewer ZirC a lot for pointing out the related diffusion-based methods, which can make our evaluation more comprehensive.
> We conduct a comparison of Text-diffuser [1] and SDXL [2] on our test dataset in both style-free and style-cond settings. We did not include DiffUTE [3] as the authors did not release the model and training dataset. The quantitative results are presented in [Table 1](https://anonymous.4open.science/r/DiffSTE-TMLR-rebuttal-8FC7/pdfs/compare-model-ocrtable.pdf) and example generations are listed in [Figure 1](https://anonymous.4open.science/r/DiffSTE-TMLR-rebuttal-8FC7/pdfs/compare-model-style-free.pdf) and [Figure 2](https://anonymous.4open.science/r/DiffSTE-TMLR-rebuttal-8FC7/pdfs/compare-model-style-cond.pdf).
>
> We highlight the following observations:
>
> **1. Significant improvements over SDXL.** Compared to the backbone model we use, Stable-diffusion-v1-5-inpainting (SD) (865M), SDXL has much more than twice the parameters(2.7B). Nevertheless, the proposed dual-encoder design and instruction tuning framework can significantly improve the scene text editing capability of SD. Equipped with our method, the SD model outperforms the much larger model, SDXL, with ~70\% absolute improvement in text correctness rate.
>
> **2. Comparable performance as TextDiffuser with less input information**. Our method achieves state-of-the-art performance and is comparable to TextDiffuser. This is impressive, especially considering TextDiffuser has an advantage in the input. Particularly, TextDiffuser first renders a layout image that already contains the ground-truth content and other information about the scene text to be added or edited. After that, the diffusion model will take the user prompt and the layout image for a generation. In comparison, our method only relies on the user prompt but still achieves good performance on text correctness. We additionally discuss the divergence in input information in point 3 below.
>
> **3. Discussions on the problem setting divergence between TextDiffuser and our method.** In our paper, we mainly consider more flexible scene text editing, where users only need to specify the style information (e.g., color and font) and the content of texts through prompts. The diffusion model can also infer the style that is coherent with background and other parts in the image. In comparison, TextDiffuser focuses on specifying most text information via the layout image. While achieving better control of the generation, flexibility can sometimes be hurt. For example, we list two scenarios where the layout image in TextDiffuser cannot accurately and naturally specify the target style as follows:
> - **Scene text editing on curved region or perspective changes:** In the first and second columns in [Figure 1](https://anonymous.4open.science/r/DiffSTE-TMLR-rebuttal-8FC7/pdfs/compare-model-style-free.pdf) and [Figure 2](https://anonymous.4open.science/r/DiffSTE-TMLR-rebuttal-8FC7/pdfs/compare-model-style-cond.pdf), we illustrate examples where the target region is highly curved or has perspective changes. Text-diffuser cannot provide high-quality and visually appealing text layouts within such regions. This is mainly because TextDiffuser highly depends on the rendered layout image. Once the layout image for target scene texts cannot be generated perfectly (which is exactly the case in those examples), the generation quality would be largely decreased. In comparison, our method provides a more flexible and robust scene text editing for such scenarios and can generate significantly better images.
> - **Text-diffuser cannot support style editing:**  As can be seen in the fourth row of [Figure 2](https://anonymous.4open.science/r/DiffSTE-TMLR-rebuttal-8FC7/pdfs/compare-model-style-cond.pdf), textdiffuser cannot understand the style specification instructions and simply conducts style-free generation by generating texts following the color of other surrounding texts (column 1, 3, 4). In comparison, our instruction-tuned DifSTE model generates text faithful to the specified text style, as shown in the third row in [Figure 2](https://anonymous.4open.science/r/DiffSTE-TMLR-rebuttal-8FC7/pdfs/compare-model-style-cond.pdf), which offers users more flexibility in application.
>
> We have included the discussions, related figures, and tables in Section E. 3, page 30-31, in the appendix of the revised paper.
>
> [1] Chen, Jingye, et al. "TextDiffuser: Diffusion Models as Text Painters." arXiv preprint arXiv:2305.10855 (2023).
> [2] Podell, Dustin, et al. "Sdxl: Improving latent diffusion models for high-resolution image synthesis." arXiv preprint arXiv:2307.01952 (2023).
> [3] Chen, Haoxing, et al. "DiffUTE: Universal Text Editing Diffusion Model."

---

### Review · Reviewer_X8wx · 2023-10-11

**Summary Of Contributions:**

Lacking the capability of generating correct text is a well-known limitation of stable-diffusion-like text-to-image generative models. The paper proposed DiffSTE that significantly outperforms SD / SD2 in terms of text correctness rate. DiffSTE adopts two encoders (a character-level encoder and a sentence-level) to encode the instruction. Later experiments (Table 2) show that the character-level encoder is crucial in boosting the text correctness. To enrich the training data, the author generates a synthetic dataset by rendering random text with random color and fonts in images and trains DiffSTE on the combination of synthetic samples and real-world text-image samples. Experiments show that DiffSTE is better than SD/SD2/Mostel in text correctness and is on par with SD2 in image naturalness. In addition, the author shows that DiffSTE is able to perform zero-shot style combination, such as interpolate between two fonts.

**Audience:**

Yes

**Claims And Evidence:**

Yes

**Requested Changes:**

1. The author needs to conduct ablation study in synthetic v.s. real mixture ratio.
2. Is there a way to make the image more natural? Sometimes, it is okay to sacrifice a little bit of text correctness for more natural image.

**Strengths And Weaknesses:**

Strengths

1. The text correctness of DiffSTE is significantly better than SD/SD2.
2. The instruction tuning design enables DiffSTE to perform zero-shot style combination and generalization. For example, the font extrapolation results in Figure 17 is interesting.

Weaknesses

1. According to Table 2, adopting the character-level encoder is the key to rendering correct texts. However, the character-level encoder design is largely inspired by the `Character-Aware Models Improve Visual Text Rendering` paper and is not novel.
2. The author proposed to train the model on the combination of real-world dataset and synthetic dataset. The ratio for mixing the synthetic samples and real-world samples can be an important design choice. However, the author has not conducted ablation study about the mixture ratio. I think the paper just mentioned getting a 1.3M dataset that combines 600K synthetic samples and 700K real-world samples. The ablation study on synthetic v.s. real ratio is important because it can help quantify the contribution of synthetic samples for scene-text generation.
3. According to Table 1, DiffSTE is on par with SD/SD2 in terms of image naturalness. In fact, the naturalness is worse than SD2 in TextOCR and is worse than SD in ArT. Samples in Figure 4 also show that the text edits generated by DiffSTE does not align with the image style very well, and seems to align too well with the fonts. This glimpse of unnaturalness is a limitation of DiffSTE.

---

> ### Author Response · Authors · 2023-12-22
> **Response to Reviewer X8wx**
>
> We thank Reviewer X8wx for the insightful suggestions for our paper. In the reminder, we want to address the key points raised in reviews.

---

> ### Author Response · Authors · 2023-12-22
> **Q1: Novelty of character encoder.**
>
> We sincerely thank reviewer X8wx for pointing out the related work [1]. We acknowledge that both methods share a similar motivation in terms of using a character-aware encoder. Nevertheless, we would like to discuss the difference between our method and theirs as well as the novelty of our work as below.
>
> **1. Differences in problem settings and target scenarios.** First, the related work [1] aims to improve the capability of diffusion models to generate new images with text in them. To improve the correctness of generated texts in the images, they introduce a new character-aware transformer encoder. In comparison, our work considers a **new scene text editing setting** where the users leverage diffusion models to add or edit texts on existing images. To solve this distinct problem, we propose a series of techniques, including the character encoder, the instruction tuning framework, additional synthetic data for skill learning, etc. In the experiment parts, we further demonstrate the effectiveness of our method for scene text editing, which is a new problem that is not covered by [1].
>
>
> **2. Different design and findings of the character-aware encoder.** Second, although both methods propose leveraging character-aware encoders for better performance, we still make new explorations in this direction. Particularly, we adopt a different design, where the add-on character-embedding layer only contains **3.7K parameters**. In comparison, the previous methods [1] designed character-aware encoders with **220M/2.6B/9B parameters** added to the full text-to-image diffusion model. Our exploration demonstrates that a much lighter-weight character encoder can also significantly boost the scene text generation capabilities of diffusion models, leading to better efficiency and lower training costs.
>
> In summary, instead of simply leveraging the character encoder, our work proposes additional techniques to tackle the new problem with distinct challenges. Our exploration of different character encoder designs also brings useful conclusions.
>
> [1] Liu, Rosanne, et al. "Character-aware models improve visual text rendering." arXiv preprint arXiv:2212.10562 (2022).

---

> ### Author Response · Authors · 2023-12-22
> **Q2: Ablation study in synthetic v.s. Real mixture ratio.**
>
> We would like to express our gratitude for the valuable suggestion from Reviewer X8wx on the mixture ratio, which would greatly help us understand the effectiveness of our method. To comprehensively study the mixture ratio between synthetic and real-world data, we have trained two additional versions of our method. Specifically, we trained one model on synthetic data only and another on real-world text images only. We compare these two trained models to the original DiffSTE in both style-free and style-conditional settings, as described in the main paper. The OCR results are presented in [Table 1](https://anonymous.4open.science/r/DiffSTE-TMLR-rebuttal-8FC7/pdfs/ablate-data-ocrtable.pdf), and examples of the generated data can be found in [Figure 1](https://anonymous.4open.science/r/DiffSTE-TMLR-rebuttal-8FC7/pdfs/ablate-data-style-free.pdf) and [Figure 2](https://anonymous.4open.science/r/DiffSTE-TMLR-rebuttal-8FC7/pdfs/ablate-data-style-cond.pdf).
>
>
> We highlight the following observations:
>
> **1.**, all three models achieve significantly **better text correctness** compared to the vanilla stable-diffusion 1/2 model and even the fine-tuned SD1/2. This demonstrates that our dual encoder design effectively helps the base diffusion model understand correct word spelling and generate accurate visual texts.
>
> **2.**, training on **real or synthetic data only hurts the text's correctness**. The combined DiffSTE model achieves more than 20\% absolute percentage improvement.
>
> **3.** training on **synthetic images only hurts the image quality**. As can be seen in the fourth row of [Figure 1](https://anonymous.4open.science/r/DiffSTE-TMLR-rebuttal-8FC7/pdfs/ablate-data-style-free.pdf), the visual texts generated by the model trained on synthetic data only do not align well with other surrounding texts. This issue arises from the limited text layouts and styles in the synthetic dataset we use. Therefore, we include real data in DiffSTE training to provide the variability of background scenes, enrich the text styles, and enable the model to better generalize to test-time real-world text images.
>
> **4.** Finally, we also want to highlight that **synthetic data builds the foundation for the editing ability** of DiffSTE. We explicitly construct the data to teach the model to follow editing instructions and infer style from surrounding texts (see Sec 3.2 in the main paper), which is the main focus of our paper.  As can be seen in the fourth row in [Figure 2](https://anonymous.4open.science/r/DiffSTE-TMLR-rebuttal-8FC7/pdfs/ablate-data-style-cond.pdf), the model trained on real data only does not understand the style control instructions, thus the generated texts are always a plain text style, either matching the surrounding text or the background similar to style-free generation. The text correctness is also affected as the model never sees style specification instructions in training (failed generation in row 4 column 4 of [Figure 2](https://anonymous.4open.science/r/DiffSTE-TMLR-rebuttal-8FC7/pdfs/ablate-data-style-cond.pdf)).
>
> We have included the discussions, related figures, and tables in Section E. 2, page 29, in the appendix of the revised paper.

---

> ### Author Response · Authors · 2023-12-22
> **Q3: Improve the image naturalness.**
>
> We sincerely thank Reviewer X8wx for pointing out the naturalness issue, and we totally agree that the naturalness of generated images plays an important role for generative models.
>
> Overall, DiffSTE effectively **produces natural texts**, as evidenced by the human evaluations detailed in the main paper. The unnatural appearance of texts might arise from the absence of surrounding scene texts, which makes inferring the correct style quite challenging. For instance, in columns 1 and 6 of Figure 4 in the main paper, it's evident that when more surrounding texts are present, DiffSTE is better able to generate natural-looking texts.
>
> It's also important to note that issues with naturalness could be partly attributed to the low resolution (256x256) of the training images in our dataset, a limitation imposed by computational constraints. Additionally, the synthetic nature of our data, where rendered text is simply placed onto a background image, might also contribute to this unnaturalness. Enhancing the quality of our training image dataset is a potential way to improve overall image quality. We leave it for future work.

---

### Review · Reviewer_KBmj · 2023-12-08

**Summary Of Contributions:**

This work introduces a diffusion model-based text editing technique that supports style-free and style-conditioned scene text-editing task with natural language instruction.

**Audience:**

Yes

**Broader Impact Concerns:**

Generative models can be misused for spreading misinformation and strengthen existing bias in the data by its nature. Besides, this work can be misused for faking documents. The authors should at least discuss them.

**Claims And Evidence:**

Yes

**Requested Changes:**

- (optional) Add image instruction.
- (obliged) Discuss the missing related works, and compare with them if applicable.

**Strengths And Weaknesses:**

Strength:
- In this work, editing is conducted by giving natural language instructions, rather than reference images as previous works, which is more convenient.
- The performance gain over baseline methods is validated by both objective and subjective metrics.
- The paper is clearly written and easy to follow.

Weaknesses:

- In Figure 6 (actually it is a table?), the advantage over baseline methods is not significant / consistent. The authors argue that this can be explained by the difference in instruction format, for which the baseline methods use a reference image, while this work uses natural language. This is a reasonable explanation, and I appreciate the author's honesty. However, I feel the reference image is easily available in real world scenarios. If you can specify a desired text by its characters, font, and color, you can directly render it as an example. Using a reference image as an extra condition is very likely to make this method more appealing.
- Some recent works [1-2] are not covered.

[1] TextStyleBrush: Transfer of Text Aesthetics From a Single Example

[2] Scene Style Text Editing

---

> ### Author Response · Authors · 2023-12-22
> **Response to Reviewer KBmj**
>
> We thank Reviewer KBmj for the insightful suggestions for our paper. In the reminder, we want to address the key points raised in reviews.

---

> > ### Author Response · Authors · 2023-12-22
> > **Q1: Reference image is easily available in real world scenarios**
> >
> > We thank Reviewer KBmj for the insightful question on the design choice of including reference images
> > for better image generation performance. We would like to discuss this point via analyses and extensive
> > experiments as below:
> >
> > **1. Including reference images v.s. instruction only.**
> > We would like to first discuss why we rely on instruction only in our method. We fully agree with your opinion that rendering a reference image could be convenient and easily available in real-world scenarios. When we design our method, our overall objective is to improve the scene-text-editing ability of diffusion models with **the minimum cost required and inference flexibility**. If users would like to do scene text editing with their local model, simply providing a prompt would be the most convenient choice for the user, as opposed to rendering a new reference image by themselves. Therefore, we propose a simple instruction-based editing framework and our experiment shows that describing the text and styles via instructions is sufficient for the diffusion model to generate high-quality images with scene texts.
> >
> > Regarding your suggestion of including the reference image as an additional condition for diffusion models, we discuss it at point 3 below with a new diffusion model that incorporates reference images.
> >
> > **2.Insufficient utilization of reference images in baselines.**
> > We agree that the GAN baselines (SRNet, Mostel) sometimes achieve better performance in terms of font and color accuracy with the aid of a reference image. However, their lack of naturalness (as indicated in Table 1 and Table 2 in main paper) significantly hinders their usability in realistic scenarios. This issue arises because GAN baselines utilize the ground-truth text clip as the style reference image. Yet, we can only render a style reference that does not fit the background image for editing style or adding new text, as there is no ground-truth style reference available.
> >
> > **3. Make our method compatible with reference images for better performance.**
> >  We would like to express our gratitude for your valuable suggestions of using additional reference images as extra conditions, which should significantly improve the performance. Indeed, we have an ongoing work that adopts a similar idea as you suggested. We deliver a scene text editing diffusion model that can either **rely on prompt only** or **use both prompt and reference image**. The design of the reference image is exactly as you suggested, where we first render the text with the desired font and color to serve as the style reference for the model. We show some visualizations for demonstration in [Figure 1](https://anonymous.4open.science/r/DiffSTE-TMLR-rebuttal-8FC7/pdfs/compare-refimage.pdf). We want to highlight that both methods can successfully control the text style, either via instruction or from the style reference image. But we do agree that a reference image might be more useful if it is more difficult for the user to specify the color name or font name in the instruction.
> >
> > We have included the discussions, related figures, and tables in Section E.1, page 28, in the appendix of the revised paper.

---

> > > ### Comment · Reviewer_KBmj · 2023-12-30
> > >
> > > The analysis and result seems reasonable to me.

---

> ### Author Response · Authors · 2023-12-22
> **Q2: Image instructions.**
>
> As described in Section 3.2 synthetic data generation. For style-free text generations, the instructions are always *Write “TEXT”*, where “TEXT” is the target word. For style text generations, the instructions are *Write “TEXT” in color: "COLOR" and font: "FONT"*, where “TEXT” is the target word, "COLOR" is the name of the target color and "FONT" is the name for target font. For baseline methods that require language instructions, i.e., pre-trained stable-diffusion models, the instruction format is *A text “TEXT”*, as we empirically find that including the verb ‘write’ hurts the OCR correctness for pre-trained stable-diffusion models.

---

> ### Author Response · Authors · 2023-12-22
> **Q3: Some recent works are not covered.**
>
> We thank reviewer KBmj for mentioning the missing related works; we have included the discussion in the related work section of the revised paper. Below are the discussions for the mentioned TextStyleBrush [1] and SSTE [2].
>
> *To reduce the dependence of paired synthetic data with source style images, and target style images, TextstyleBrush [1] proposes to disentangle the text appearance into content and style vectors in a self-supervised manner, which allows the utilization of real-world text images.
> Another recent work, SSTE [2], proposes to embed the text styles in the latent feature space, thus allowing users to control the text style, such as text rotation, text color, and font via latent space editing similar to StyleGAN [3].*
>
> [1] Krishnan, Praveen, et al. "Textstylebrush: Transfer of text aesthetics from a single example." IEEE Transactions on Pattern Analysis and Machine Intelligence (2023).
>
> [2] Su, Tonghua, et al. "Scene Style Text Editing." arXiv preprint arXiv:2304.10097 (2023).
>
> [3] Karras, Tero, Samuli Laine, and Timo Aila. "A style-based generator architecture for generative adversarial networks." Proceedings of the IEEE/CVF conference on computer vision and pattern recognition. 2019.

---

> ### Author Response · Authors · 2023-12-22
> **Q4: Missing discussion of potential risks.**
>
> We sincerely thank Reviewer KBmj for mentioning the potential risks of our method. We have included the discussion of broader impacts in Section F, page 32, in the appendix of our revised paper. Below are the discussions:
>
> *In this paper, we propose a novel method DiffSTE to adapt pre-trained diffusion models for scene text editing. Our proposed method consists of an instruction encoder for better style control and a character encoder for improved text legibility. DiffSTE improves text correctness for the pre-trained diffusion model and further allows users to control the generated text styles. However, we also admit that our method can be used to forge signatures or spread misinformation, which might have a significant impact on society. In practice, our method should be appropriately used with careful checks on potential risks. For example, we can adopt recently proposed diffusion-watermarking methods [1,2,3] to identify whether certain text images are generated by our model to identify forged signatures.*
>
> [1] Zhao, Yunqing, et al. "A recipe for watermarking diffusion models." arXiv preprint arXiv:2303.10137 (2023).
> [2] Liu, Yugeng, et al. "Watermarking Diffusion Model." arXiv preprint arXiv:2305.12502 (2023).
> [3] Wen, Yuxin, et al. "Tree-Ring Watermarks: Fingerprints for Diffusion Images that are Invisible and Robust." arXiv preprint arXiv:2305.20030 (2023).

---

### Decision · Action_Editor_jgeT · 2024-02-06

**Recommendation:** Accept as is

**Comment:**

Lacking the capability of  is a well-known limitation of stable-diffusion-like text-to-image generative models. The paper proposed DiffSTE for generating correct text, which is shown to significantly outperform SD / SD2 in terms of text correctness rate. DiffSTE adopts two encoders (a character-level encoder and a sentence-level) to encode the instruction. Experiments also show that the character-level encoder is crucial in boosting the text correctness. The authors proposed a synthetic dataset by rendering random text with random color and fonts in images and trains DiffSTE on the combination of synthetic samples and real-world text-image samples. In addition, the author shows that DiffSTE is able to perform zero-shot style combination, such as interpolate between two fonts.

After reading author's responses, all reviewers are leaning to accept this paper.

**Audience:**

Yes.

**Claims And Evidence:**

Yes.